# Human EEG and artificial neural networks reveal disentangled representations and processing timelines of object real-world size and depth in natural images

**Zitong Lu[1,2]\*, Julie Golomb[1]**

[1]Department of Psychology, The Ohio State University, Columbus, United States; [2]McGovern Institute for Brain Research, Massachusetts Institute of Technology, Boston, United States

## eLife Assessment

This **important** study combines EEG, neural networks and multivariate pattern analysis to show that real-world size, retinal size and real-world depth are represented at different latencies. The evidence presented is **convincing** and the work will be of broader interest to the experimental and computational vision community.

**\*For correspondence:**
zitonglu@mit.edu

**Abstract** Remarkably, human brains have the ability to accurately perceive and process the real-world size of objects, despite vast differences in distance and perspective. While previous studies have delved into this phenomenon, distinguishing the processing of real-world size from other visual properties, like depth, has been challenging. Using the THINGS EEG2 dataset with human EEG recordings and more ecologically valid naturalistic stimuli, our study combines human EEG and representational similarity analysis to disentangle neural representations of object real-world size from retinal size and perceived depth, leveraging recent datasets and modeling approaches to address challenges not fully resolved in previous work. We report a representational timeline of visual object processing: object real-world depth processed first, then retinal size, and finally, real-world size. Additionally, we input both these naturalistic images and object-only images without natural background into artificial neural networks. Consistent with the human EEG findings, we also successfully disentangled representation of object real-world size from retinal size and real-world depth in all three types of artificial neural networks (visual-only ResNet, visual-language CLIP, and language-only Word2Vec). Moreover, our multi-modal representational comparison framework across human EEG and artificial neural networks reveals real-world size as a stable and higher-level dimension in object space incorporating both visual and semantic information. Our research provides a temporally resolved characterization of how certain key object properties – such as object real-world size, depth, and retinal size – are represented in the brain, which offers further advances and insights into our understanding of object space and the construction of more brain-like visual models.

## Introduction

Imagine you are viewing an apple tree while walking around an orchard: as you change your perspective and distance, the retinal size of the apple you plan to pick varies, but you still perceive the apple as having a constant real-world size. How do our brains extract object real-world size information

during object recognition to allow us to understand the complex world? Behavioral studies have demonstrated that perceived real-world size is represented as an object physical property, revealing same-size priming effects (*Setti et al., 2009*), familiar-size stroop effects (*Konkle and Oliva, 2012a*; *Long and Konkle, 2017*), and canonical visual size effects (*Chen et al., 2022*; *Konkle and Oliva, 2011*). Human neuroimaging studies have also found evidence of object real-world size representation (*Huang et al., 2022*; *Khaligh-Razavi et al., 2018*; *Konkle and Caramazza, 2013*; *Konkle and Oliva, 2012b*; *Luo et al., 2023*; *Quek et al., 2023*; *Wang et al., 2022a*). These findings suggest real-world size is a fundamental dimension of object representation.

However, previous studies on object real-world size have faced several challenges. Firstly, the perception of an object's real-world size is closely related to the perception of its real-world distance in depth. For instance, imagine you are looking at photos of an apple and a basketball: if the two photos were zoomed in such that the apple and the basketball filled the same exact retinal (image) size, you could still easily perceive that the apple is the physically smaller real-world object. But you would simultaneously infer that the apple is thus located closer to you (or the camera) than the basketball. In previous neuroimaging studies of perceived real-world size (*Huang et al., 2022*; *Konkle and Caramazza, 2013*; *Konkle and Oliva, 2012b*), researchers presented images of familiar objects zoomed and cropped such that they occupied the same retinal size, finding that neural responses in ventral temporal cortex reflected the perceived real-world size (e.g. an apple smaller than a car). However, while they controlled the retinal size of objects, the intrinsic correlation between real-world size and real-world depth in these images meant that the influence of perceived real-world depth could not be entirely isolated when examining the effects of real-world size. This makes it difficult to ascertain whether their results were driven by neural representations of perceived real-world size and/or perceived real-world depth. MEG and EEG studies focused on temporal processing of object size representations *Khaligh-Razavi et al., 2018*; *Wang et al., 2022a* have been similarly susceptible to this limitation. Indeed, one recent study (*Quek et al., 2023*) provided evidence that perceived real-world depth could influence real-world size representations, further illustrating the necessity of investigating pure real-world size representations in the brain. Secondly, the stimuli used in these studies were cropped object stimuli against a plain white or gray background, which are not particularly naturalistic. More and more studies and datasets have highlighted the important role of naturalistic context in object recognition (*Allen et al., 2022*; *Gifford et al., 2022*; *Grootswagers et al., 2022*; *Hebart et al., 2019*; *Stoinski et al., 2024*). In ecological contexts, inferring the real-world size/distance of an object likely relies on a combination of bottom-up visual information and top-down knowledge about canonical object sizes for familiar objects. Incorporating naturalistic background context in experimental stimuli may produce more accurate assessments of the relative influences of visual shape representations (*Bracci et al., 2017*; *Bracci and Op de Beeck, 2016*; *Proklova et al., 2016*) and higher-level semantic information (*Doerig et al., 2022*; *Huth et al., 2012*; *Wang et al., 2022b*). Furthermore, most previous studies have tended to categorize size rather broadly, such as merely differentiating between big and small objects (*Khaligh-Razavi et al., 2018*; *Konkle and Oliva, 2012b*; *Wang et al., 2022a*) or dividing object size into seven levels from small to big. To more finely investigate the representation of object size in the brain, it may be necessary to obtain a more continuous measure of size for a more detailed characterization.

Certainly, a minority of fMRI studies have attempted to utilize natural images and also engaged in more detailed size measurements to more precisely explore the encoding of object real-world size in different brain areas (*Luo et al., 2023*; *Troiani et al., 2014*). However, no study has yet comprehensively overcome all the challenges and unfolded a clear processing timeline for object retinal size, real-world size, and real-world depth in human visual perception.

In the current study, we overcome these challenges by combining human EEG recordings, naturalistic stimulus images, artificial neural networks, and computational modeling approaches including representational similarity analysis (RSA) and partial correlation analysis to distinguish the neural representations of object real-world size, retinal size, and real-world depth. We applied our integrated computational approach to an open EEG dataset, THINGS EEG2 (*Gifford et al., 2022*). Firstly, the visual image stimuli used in this dataset are more naturalistic and include objects that vary in real-world size, depth, and retinal size. This allows us to employ a multi-model representational similarity analysis to investigate relatively unconfounded representations of object real-world size, partialing out – and simultaneously exploring – these confounding features. Secondly, we are able to explore the

neural dynamics of object feature processing in a more ecological context based on natural images in human object recognition. Thirdly, instead of categorizing object size into discrete levels, we applied a more continuous measure based on detailed behavioral measurements from an online size rating task, allowing us to more finely decode the representation of object size in the brain.

We first focus on unfolding the neural dynamics of statistically isolated object real-world size representations. The temporal resolution of EEG allows us the opportunity to investigate the representational time course of visual object processing, asking whether processing of perceived object real-world size precedes or follows processing of perceived depth, if these two properties are in fact processed independently.

We then attempt to further explore the underlying mechanisms of how human brains process object size and depth in natural images by integrating artificial neural networks (ANNs). In the domain of cognitive computational neuroscience, ANNs offer a complementary tool to study visual object recognition, and an increasing number of studies support that ANNs exhibit representations similar to human visual systems (*Cichy et al., 2016*; *Güçlü and van Gerven, 2015*; *Yamins et al., 2014*; *Yamins and DiCarlo, 2016*). Indeed, a recent study found that ANNs also represent real-world size (*Huang et al., 2022*); however, their use of a fixed retinal size image dataset with the same cropped objects as described above makes it similarly challenging to ascertain whether the results reflected real-world size and/or depth. Additionally, some recent work indicates that artificial neural networks incorporating semantic embedding and multimodal neural components might more accurately reflect human visual representations within visual areas and even the hippocampus, compared to vision-only networks (*Choksi et al., 2022a*; *Choksi et al., 2022b*; *Conwell et al., 2022*; *Doerig et al., 2022*; *Jozwik et al., 2023*; *Wang et al., 2022b*). Given that perception of real-world size may incorporate

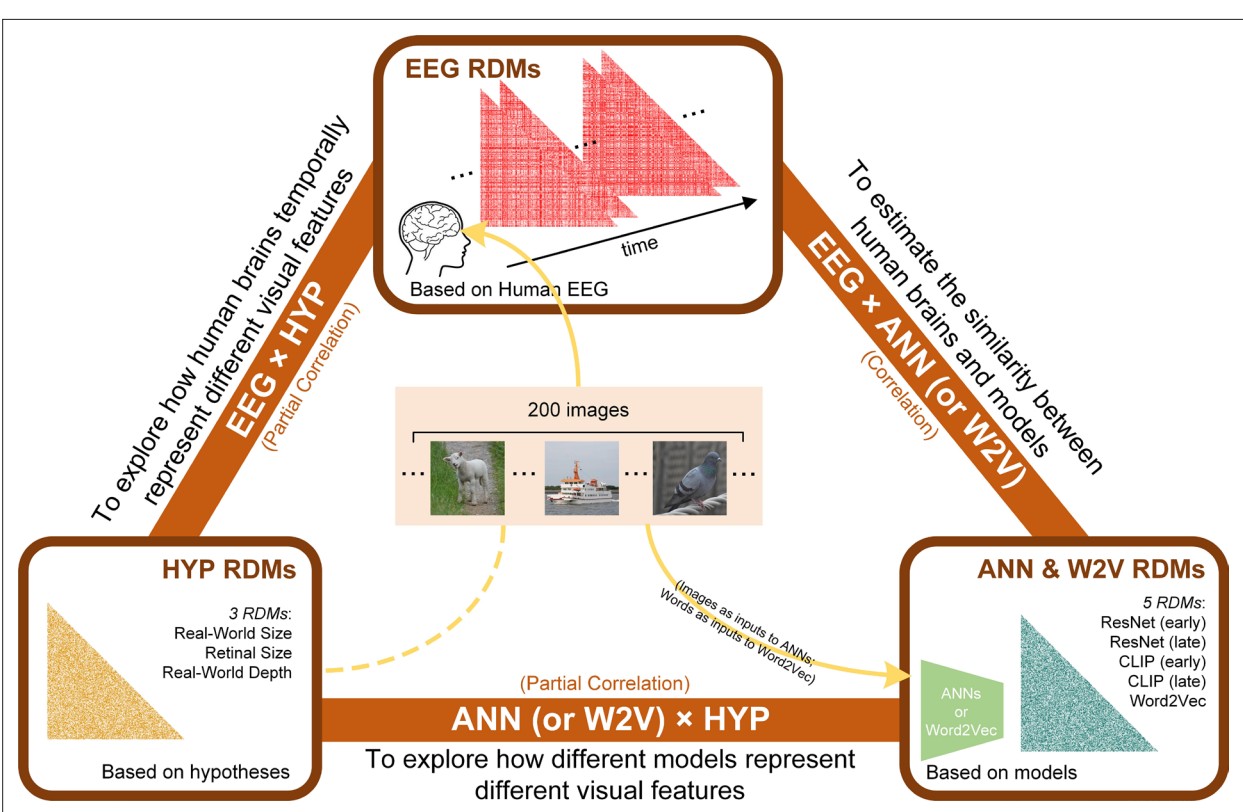

**Figure 1.** Overview of our analysis pipeline including constructing three types of RDMs and conducting comparisons between them. We computed RDMs from three sources: neural data (EEG), hypothesized object features (real-world size, retinal size, and real-world depth), and artificial models (ResNet, CLIP, and Word2Vec). Then we conducted cross-modal representational similarity analyses between: EEG ×HYP (partial correlation, controlling for other two HYP features), ANN (or W2V) ×HYP (partial correlation, controlling for other two HYP features), and EEG ×ANN (correlation).

The online version of this article includes the following figure supplement(s) for figure 1:

**Figure supplement 1.** Four ANN RDMs of ResNet early layer, ResNet late layer, CLIP early layer, and CLIP late layer.

**Figure supplement 2.** Word2Vec RDM.

both bottom-up visual and top-down semantic knowledge about familiar objects, these models offer yet another novel opportunity to investigate this question. Utilizing both visual and visual-semantic models, as well as different layers within these models, ANNs provide us the approach to extract various image features, low-level visual information from early layers, and higher-level information including both visual and semantic features from late layers.

The integrated computational approach by cross-modal representational comparisons we take with the current study allows us to compare how representations of perceived real-world size and depth emerge in both human brains and artificial neural networks. Unraveling the internal representations of object size and depth features in both human brains and ANNs enables us to investigate how distinct spatial properties—retinal size, real-world depth, and real-world size—are encoded across systems, and to uncover the representational mechanisms and temporal dynamics through which real-world size emerges as a potentially higher-level, semantically grounded feature.

## Results

We conducted a cross-modal representational similarity analysis (*Figures 1 and 2*, see Materials and methods section for details) comparing the patterns of human brain activation while participants viewed naturalistic object images (timepoint-by-timepoint decoding of EEG data), the output of different layers of artificial neural networks and semantic language models fed the same stimuli (ANN and Word2Vec models), and hypothetical patterns of representational similarity based on behavioral and mathematical measurements of different visual image properties (perceived real-world object size, displayed retinal object size, and inferred real-world object depth).

### Dynamic representations of object size and depth in human brains

To explore if and when human brains contain distinct representations of perceived real-world size, retinal size, and real-world depth, we constructed timepoint-by-timepoint EEG neural RDMs (*Figure 2A*), and compared these to three hypothesis-based RDMs corresponding to different visual image properties (*Figure 2B*). Firstly, we confirmed that the hypothesis-based RDMs were indeed correlated with each other (*Figure 3A*), and without accounting for the confounding variables, Spearman correlations between the EEG and each hypothesis-based RDM revealed overlapping periods of representational similarity (*Figure 3B*). In particular, representational similarity with real-world size (from 90 to 120ms and from 170 to 240ms) overlapped with the significant time windows of other features, including retinal size from 70 to 210ms, and real-world depth from 60 to 130ms and from 180 to 230ms. But critically, with the partial correlations, we isolated their independent representations. The partial correlation results reveal a relatively unconfounded representation of object real-world size in the human brain from 170 to 240ms after stimulus onset, independent from retinal size and real-world depth, which showed significant representational similarity at different time windows (retinal size from 90 to 200ms, and real-world depth from 60 to 130ms and 270–300ms; *Figure 3D*).

Peak latency results showed that neural representations of real-world size, retinal size, and real-world depth reached their peaks at different latencies after stimulus onset (real-world depth: ~87ms, retinal size: ~138ms, real-world size: ~206ms, *Figure 3C*). The representation of real-world size had a significantly later peak latency than that of both retinal size, $t(9)=4.30$, $p=0.002$, and real-world depth, $t(9)=18.58$, $p<0.001$. And retinal size representation had a significantly later peak latency than real-world depth, $t(9)=3.72$, $p=0.005$. These varying peak latencies imply an encoding order for distinct visual features, transitioning from real-world depth through retinal size, and then to real-world size.

### Artificial neural networks also reflect distinct representations of object size and depth

To test how ANNs process these visual properties, we input the same stimulus images into ANN models and got their latent features from early and late layers (*Figure 2C*), and then conducted comparisons between the ANN RDMs and hypothesis-based RDMs. Parallel to our findings of dissociable representations of real-world size, retinal size, and real-world depth in the human brain signal, we also found dissociable representations of these visual features in ANNs (*Figure 3E*). Our partial correlation RSA analysis showed that early layers of both ResNet and CLIP had significant real-world depth and retinal size representations, whereas the late layers of both ANNs were dominated by

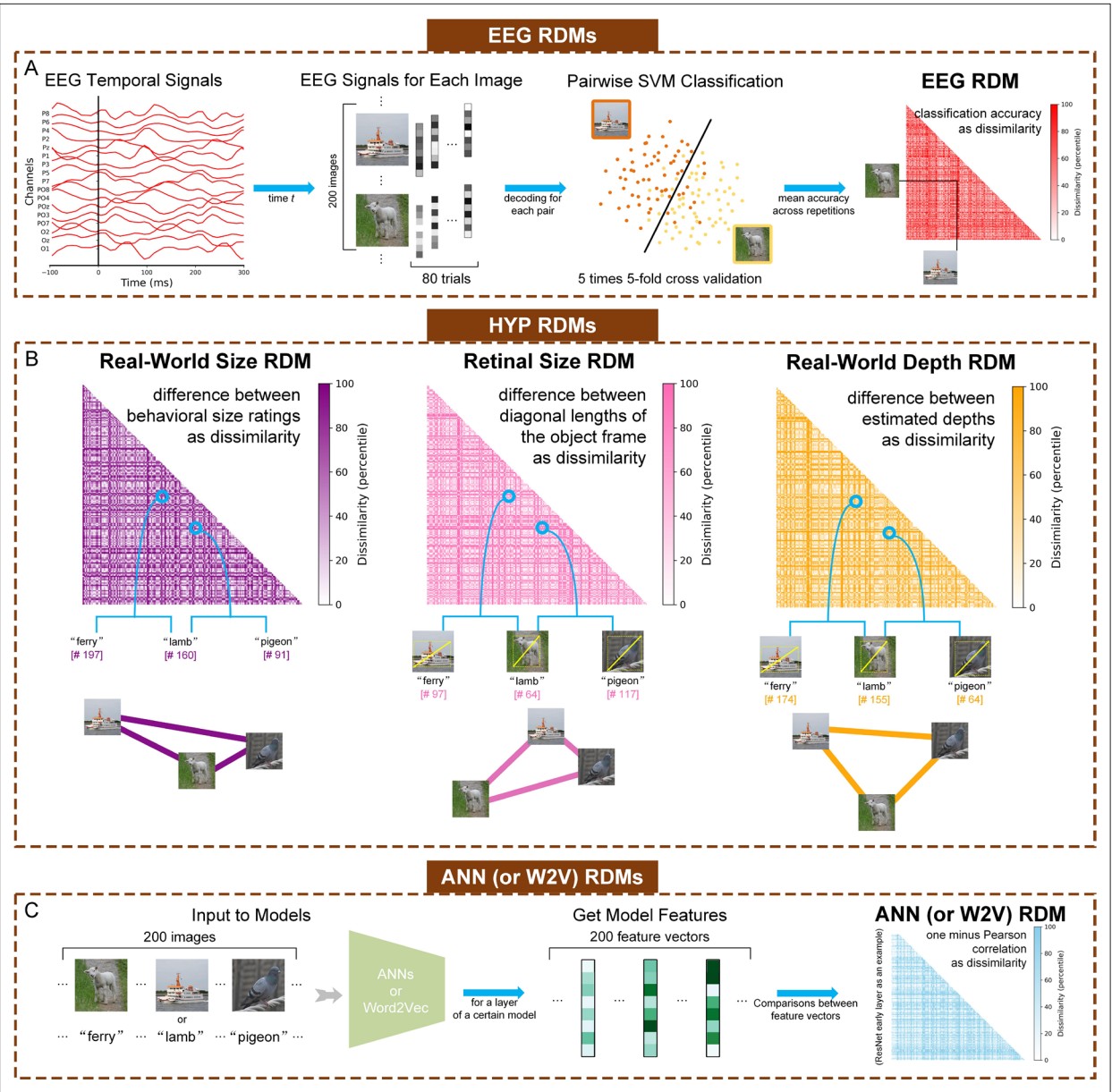

**Figure 2.** Methods for calculating neural (EEG), hypothesis-based (HYP), and artificial neural network (ANN) & semantic language processing (Word2Vec, W2V) model-based representational dissimilarity matrices (RDMs). (**A**) Steps of computing the neural RDMs from EEG data. EEG analyses were performed in a time-resolved manner on 17 channels as features. For each time t, we conducted pairwise cross-validated SVM classification. The classification accuracy values across different image pairs resulted in each 200×200 RDM for each time point. (**B**) Calculating the three hypothesis-based RDMs: Real-World Size RDM, Retinal Size RDM, and Real-World Depth RDM. Real-world size, retinal size, and real-world depth were calculated for the object in each of the 200 stimulus images. The number in the bracket represents the rank (out of 200, in ascending order) based on each feature corresponding to the object in each stimulus image (e.g. 'ferry' ranks 197th in real-world size from small to big out of 200 objects). The connection graph to the right of each RDM represents the relative representational distance of three stimuli in the corresponding feature space. (**C**) Steps of computing the ANN and Word2Vec RDMs. For ANNs, the inputs were the resized images, and for Word2Vec, the inputs were the words of object concepts. For clearer visualization, the shown RDMs were separately histogram-equalized (percentile units).

real-world size representations, though there was also weaker retinal size representation in the late layer of ResNet and real-world depth representation in the late layer of CLIP (additional results of the extended analysis of multiple layers in ResNet and CLIP are shown in *Figure 3—figure supplement 1*). The detailed statistical results are shown in *Supplementary file 1, table A*.

Thus, ANNs provide another approach to understand the formation of different visual features, offering convergent results with the EEG representational analysis, where retinal size was reflected

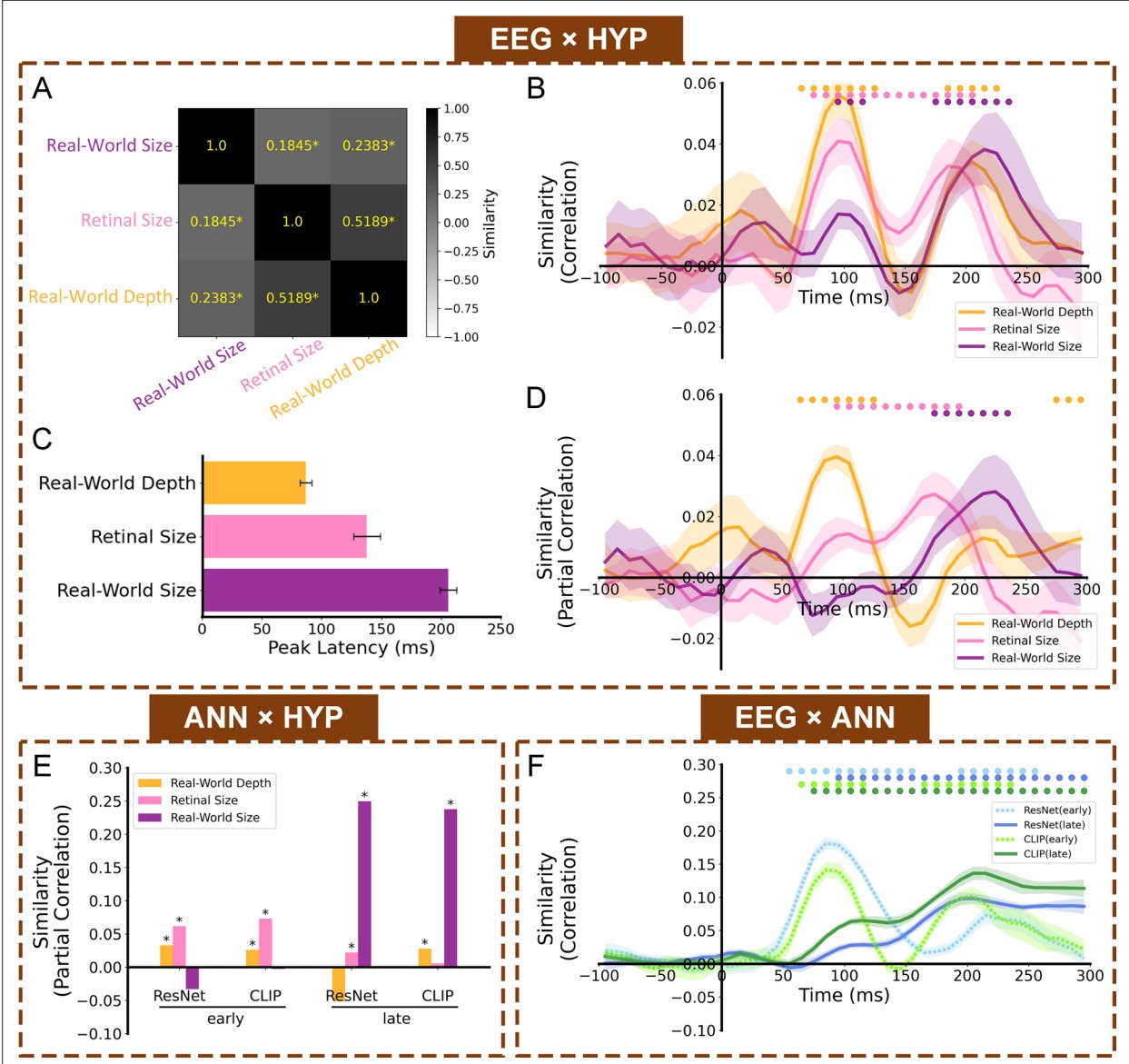

**Figure 3.** Cross-modal RSA results. (**A**) Similarities (Spearman correlations) between three hypothesis-based RDMs. Asterisks indicate a significant similarity, p<0.05. (**B**) Representational similarity time courses (full Spearman correlations) between EEG neural RDMs and hypothesis-based RDMs. (**C**) Temporal latencies for peak similarity (partial Spearman correlations) between EEG and the three types of object information. Error bars indicate ± SEM. Asterisks indicate significant differences across conditions (p<0.05); (**D**) Representational similarity time courses (partial Spearman correlations) between EEG neural RDMs and hypothesis-based RDMs. (**E**) Representational similarities (partial Spearman correlations) between the four ANN RDMs and hypothesis-based RDMs of real-world depth, retinal size, and real-world size. Asterisks indicate significant partial correlations (bootstrap test, p<0.05). (**F**) Representational similarity time courses (Spearman correlations) between EEG neural RDMs and ANN RDMs. Color-coded small dots at the top indicate significant timepoints (cluster-based permutation test, p<0.05). Shaded area reflects ± SEM across the 10 subjects.

The online version of this article includes the following figure supplement(s) for figure 3:

**Figure supplement 1.** Representational similarities (partial Spearman correlations) between ANN RDMs from multiple layers in ResNet and CLIP and hypothesis-based RDMs of real-world depth, retinal size, and real-world size (as extended results of *Figure 3E*).

**Figure supplement 2.** Representational similarity time courses (Spearman correlations) between EEG neural RDMs and ANN RDMs from multiple layers (as extended results of *Figure 3F*).

**Figure supplement 3.** Representational similarity time courses (partial Spearman correlations) between EEG neural RDMs and ANN RDMs controlling for the three hypothesis-based RDMs.

**Figure supplement 4.** Noise ceiling of representational similarity analysis based on EEG neural RDMs.

**Figure supplement 5.** RSA results based on CORnet.

most in the early layers of ANNs, while object real-world size representations didn't emerge until late layers of ANNs, consistent with a potential role of higher-level visual information, such as the semantic information of object concepts.

## Representational similarity between human EEG and artificial neural networks

To directly examine the representational similarity between ANNs and human EEG signals, we compared the timepoint-by-timepoint EEG neural RDMs and the ANN RDMs. This analysis allowed us to assess how different stages of visual processing in the human brain align temporally with hierarchical representations in ANNs. As shown in *Figure 3F*, the early layer representations of both ResNet and CLIP (ResNet.maxpool layer and CLIP.visual.avgpool) showed significant correlations with early EEG time windows (early layer of ResNet: 40–280ms, early layer of CLIP: 50–130ms and 160–260ms), while the late layers (ResNet.avgpool layer and CLIP.visual.attnpool layer) showed correlations extending into later time windows (late layer of ResNet: 80–300ms, late layer of CLIP: 70–300ms). Although there is substantial temporal overlap between early and late model layers, the overall pattern suggests a rough correspondence between model hierarchy and neural processing stages.

We further extended this analysis across intermediate layers of both ResNet and CLIP models (from early to late, ResNet: ResNet.maxpool, ResNet.layer1, ResNet.layer2, ResNet.layer3, ResNet.layer4, ResNet.avgpool; from early to late, CLIP: CLIP.visual.avgpool, CLIP.visual.layer1, CLIP.visual.layer2, CLIP.visual.layer3, CLIP.visual.layer4, CLIP.visual.attnpool). The results, now included in *Figure 3—figure supplement 2*, show a consistent trend: early layers exhibit higher similarity to early EEG time points, and deeper layers show increased similarity to later EEG stages. This pattern of early-to-late correspondence aligns with previous findings that convolutional neural networks exhibit similar hierarchical representations to those in the brain visual cortex (*Cichy et al., 2016*; *Güçlü and van Gerven, 2015*; *Kietzmann et al., 2019*; *Yamins and DiCarlo, 2016*): that both the early stage of brain processing and the early layer of the ANN encode lower-level visual information, while the late stage of the brain and the late layer of the ANN encode higher-level visual information. Notably, early brain responses showed stronger similarity to early ResNet layers than to CLIP layers, consistent with prior work suggesting that early visual processing is more closely aligned with purely visual models (*Greene and Hansen, 2020*). In contrast, at later time windows, brain activity more closely resembled late CLIP layers, possibly reflecting the integration of visual and semantic information. However, it is also possible that these differences between ResNet and CLIP reflect differences in training data scale and domain.

To contextualize how much of the shared variance between EEG and ANN representations is driven by the specific visual object features we tested above, we conducted a partial correlation analysis between EEG RDMs and ANN RDMs controlling for the three hypothesis-based RDMs (*Figure 3—figure supplement 3*). The EEG ×ANN similarity results remained largely unchanged, suggesting that ANN representations capture much more additional rich representational structure beyond these features. Similarly, a supplemental EEG noise ceiling analysis (*Figure 3—figure supplement 4*) shows that the observed EEG–HYP similarity values are substantially below a theoretical upper bound of explainable variance. This outcome is fully expected: Each of our HYP RDMs captures only a specific aspect of the neural representational structure, rather than attempting to account for the totality of the EEG or ANN signal. Our goal is not to optimize model performance or maximize fit, but to probe if these different components of object information are reflected – and dissociated – in the temporal dynamics and processing stages of the brain's responses. In parallel, these supplemental findings help to situate our hypothesis-driven analysis within the broader representational capacity of the brain and the models.

## Real-world size as a stable and higher-level dimension in object space

An important aspect of the current study is the use of naturalistic visual images as stimuli, in which objects were presented in their natural contexts, as opposed to cropped images of objects without backgrounds. In natural images, background can play an important role in object perception. How dependent are the above results on the presence of naturalistic background context? To investigate how image context influences object size and depth representations, we next applied a reverse engineering method, feeding the ANNs with modified versions of the stimulus images containing

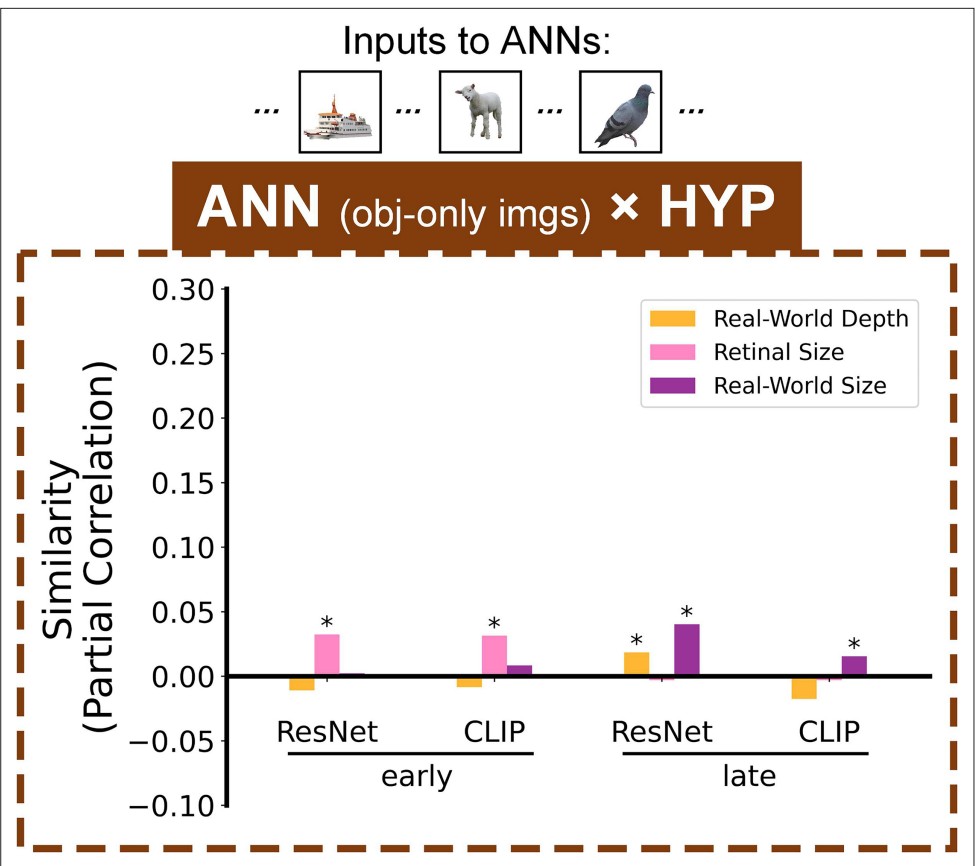

**Figure 4.** Contribution of image backgrounds to object size and depth representations. Representational similarity results (partial Spearman correlations) between ANNs fed inputs of cropped object images without backgrounds and the hypothesis-based RDMs. Stars above bars indicate significant partial correlations (bootstrap test, p<0.05).

The online version of this article includes the following figure supplement(s) for figure 4:

**Figure supplement 1.** Four ANN RDMs with inputs of cropped object images without background of ResNet early layer, ResNet late layer, CLIP early layer, and CLIP late layer.

cropped objects without background, and evaluating the ensuing ANN representations compared to the same original hypothesis-based RDMs. If the background significantly contributes to the formation of certain feature representations, we may see some encoding patterns in ANNs disappear when the input only includes the pure object but no background.

Compared to results based on images with background, the ANNs based on cropped-object modified images showed weaker overall representational similarity for all features (*Figure 4*). In the early layers of both ANNs, we now only observed significantly preserved retinal size representations (which is a nice validity check, since retinal size measurements were based purely on the physical object dimensions in the image, independent of the background). Real-world depth representations were almost totally eliminated, with only a small effect in the late layer of ResNet. However, we still observed a preserved pattern of real-world size representations, with significant representational similarity in the late layers of both ResNet and CLIP, and not in the early layers. The detailed statistical results are shown in *Supplementary file 1, table B*. Even though the magnitude of representational similarity for object real-world size decreased when we removed the background, this high-level representation was not entirely eliminated. This finding suggests that background information does indeed influence object processing, but the representation of real-world size seems to be a relatively stable higher-level feature. On the other hand, representational formats of real-world depth changed when the input lacked background information. The deficiency of real-world depth representations in early layers, compared to when using full-background images, might suggest that the human brain typically uses background information to estimate object depth, though the significant effect in the

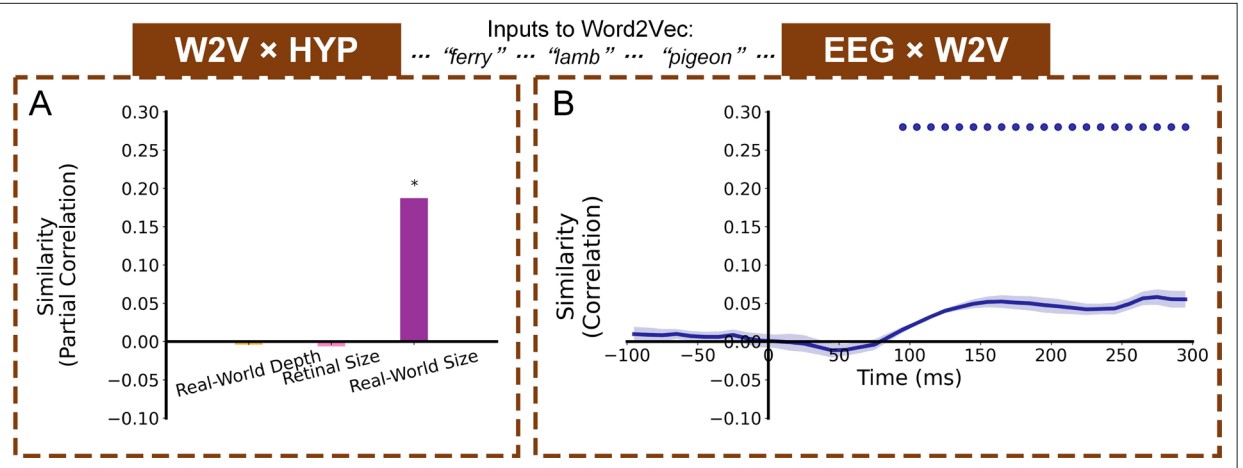

**Figure 5.** Representation similarity with a non-visual semantic language processing model (Word2Vec) fed word inputs corresponding to the images' object concepts. (**A**) Representational similarity results (partial Spearman correlations) between Word2Vec RDM and hypothesis-based RDMs. Stars above bars indicate significant partial correlations (bootstrap test, p<0.05). (**B**) Representational similarity time course (Spearman correlations) between EEG RDMs (neural activity while viewing images) and Word2Vec RDM (fed corresponding word inputs). Color-coded small dots at the top indicate significant timepoints (cluster-based permutation test, p<0.05). Line width reflects ± SEM across the 10 subjects.

The online version of this article includes the following figure supplement(s) for figure 5:

**Figure supplement 1.** Representational similarity time courses (partial Spearman correlations) between EEG neural RDMs and the Word2Vec RDM controlling for four ANN RDMs (ResNet early/late and CLIP early/late layers).

**Figure supplement 2.** Unique and shared variance of real-world size and semantic information (from Word2Vec) in EEG neural representations using variance partitioning analysis while controlling retinal size and real-world depth.

late layer of ResNet in background-absent condition might also suggest that the brain (or at least ANN) has additional ability to integrate size information to infer depth when there is no background. These results show that real-world size emerges in the later layers of ANNs regardless of image background information, and – based on our prior EEG results – although we could not test object-only images in the EEG data, we hypothesize that a similar temporal profile would be observed in the brain, even for object-only images.

The relative robustness of real-world size representations – even in the absence of background – raises an important question: might real-world size be encoded as a more abstract, conceptual-level dimension in the brain's object representation space? If so, we might expect it to be driven not only by higher-level visual information, but also potentially by purely semantic information about familiar objects. To test this, we extracted object names from each image and input the object names into a Word2Vec model to obtain a Word2Vec RDM (*Figure 1—figure supplement 2*), and then conducted a partial correlation RSA comparing the Word2Vec representations with the hypothesis-based RDMs (*Figure 5A*). The results showed a significant real-world size representation (*r*=0.1871, p<0.001) but no representation of retinal size (*r*=−0.0064, p=0.8148) or real-world depth (*r*=−0.0040, p=0.7151) from Word2Vec. Also, the significant time window (90–300ms) of similarity between Word2Vec RDM and EEG RDMs (*Figure 5B*) contained the significant time window of EEG × real-world size representational similarity (*Figure 3B*).

To further probe the relationship between real-world size and semantic information, and to examine whether Word2Vec captures variances in EEG signals beyond that explained by visual models, we conducted two additional analyses. First, we performed a partial correlation between EEG RDMs and the Word2Vec RDM, while regressing out four ANN RDMs (early and late layers of both ResNet and CLIP) (*Figure 5—figure supplement 1*). We found that semantic similarity remained significantly correlated with EEG signals across sustained time windows (100–190ms and 250–300ms), indicating that Word2Vec captures neural variance not fully explained by visual or visual-language models. Second, we conducted a variance partitioning analysis, in which we decomposed the variance in EEG RDMs explained by three hypothesis-based RDMs and the semantic RDM (Word2Vec RDM), and we still found that real-world size explained unique variance in EEG even after accounting for semantic similarity (*Figure 5—figure supplement 2*). And we also observed a substantial shared variance jointly

explained by real-world size and semantic similarity and a unique variance of semantic information. These results suggest that real-world size is indeed partially semantic in nature, but also has independent neural representation not fully explained by general semantic similarity.

Both the reverse engineering manipulation and Word2Vec findings corroborate that object real-world size representation, unlike retinal size and real-world depth, emerges in both image- and semantic-level in object space.

## Discussion

Our study applied computational methods to distinguish the representations of objects' perceived real-world size, retinal size, and inferred real-world depth features in both human brains and ANNs. Consistent with prior studies reporting real-world size representations (*Huang et al., 2022*; *Khaligh-Razavi et al., 2018*; *Konkle and Caramazza, 2013*; *Konkle and Oliva, 2012b*; *Luo et al., 2023*; *Quek et al., 2023*; *Wang et al., 2022a*), we found that both human brains and ANNs contain significant information about real-world size. Critically, our study goes beyond the contributions of prior studies in several key ways, offering both theoretical and methodological advances: (a) we eliminated the confounding impact of perceived real-world depth (in addition to retinal size) on the real-world size representation; (b) we conducted analyses based on more ecologically valid naturalistic images; (c) we obtained precise feature values for each object in every image, instead of simply dividing objects into two or seven coarse categories; and (d) we utilized a multi-modal, partial correlation RSA that combines EEG, hypothesis-based models, and ANNs. Our approach allowed us to investigate representational time courses and reverse engineering manipulations in unparalleled detail. By integrating EEG data with hypothesis-based models and ANNs, this method offers a powerful tool for dissecting the neural underpinnings of object size and depth perception in more ecological contexts, which enriches our comprehension of the brain's representational mechanisms.

Using EEG, we uncovered a representational timeline for visual object processing, with object real-world depth information represented first, followed by retinal size, and finally real-world size. While size and depth are highly correlated to each other, our results suggest that the human brain indeed has dissociated time courses and mechanisms to process them. The later representation time window for object real-world size may suggest that the brain requires more sophisticated, higher-level information to form this representation, perhaps incorporating semantic and/or memory information about familiar objects, which was corroborated by our ANN and Word2Vec analyses. These findings also align with a recent fMRI study (*Luo et al., 2023*) using natural images to explore the neural selectivity for real-world size, finding that low-level visual information could hardly account for neural size preferences, although that study did not consider covariables like retinal size and real-world depth.

In contrast to the later-emerging real-world size representations, it makes sense that retinal size representations could be processed more quickly based on more fundamental, lower-level information such as shape and edge discrimination. Interestingly, real-world depth representations emerged even earlier than retinal size, suggesting that depth feature may precede real-world size processing. This early emergence might reflect the role of depth cues in facilitating object segmentation, as proposed in classical theories of vision (*Marr, 1982*), where depth helps delineate object boundaries from complex backgrounds. Additionally, there was a secondary, albeit substantially later, significant depth representation time window, which might indicate that our brains also have the ability to integrate object retinal size and higher-level real-size information to form the final representation of real-world depth. Our comparisons between human brains and artificial models and explorations on ANNs and Word2Vec offer further insights and suggest that although real-world object size and depth are closely related, object real-world size appears to be a more stable and higher-level dimension.

The concept of 'object space' in cognitive neuroscience research is crucial for understanding how various visual features of objects are represented. Historically, various visual features have been considered important dimensions in constructing object space, including animate-inanimate (*Kriegeskorte et al., 2008*; *Naselaris et al., 2012*), spikiness (*Bao et al., 2020*; *Coggan and Tong, 2023*), and physical appearance (*Edelman et al., 1998*). In this study, we focus on one particular dimension, real-world size (*Huang et al., 2022*; *Konkle and Caramazza, 2013*; *Konkle and Oliva, 2012b*). How we generate neural distinctions of different object real-world size and where this ability comes from remain uncertain. Some previous studies found that object shape rather than texture information could trigger neural size representations (*Huang et al., 2022*; *Long et al., 2016*; *Long et al., 2018*;

*Wang et al., 2022a*). Our results attempt to further advance their findings that object real-world size is a stable and higher-level dimension substantially driven by object semantics in object space.

Increasingly, research has begun to use ANNs to study the mechanisms of object recognition (*Ayzenberg et al., 2023*; *Cichy and Kaiser, 2019*; *Doerig et al., 2023*; *Kanwisher et al., 2023*). We can explore how the human brain processes information at different levels by comparing brain activity with models (*Cichy et al., 2016*; *Khaligh-Razavi and Kriegeskorte, 2014*; *Kuzovkin et al., 2018*; *Xie et al., 2020*), and we can also analyze the representation patterns of the models with some specific manipulations and infer potential processing mechanisms in the brain (*Golan et al., 2020*; *Huang et al., 2022*; *Lu and Ku, 2023*; *Xu et al., 2021*). In the current study, our comparisons between EEG signals and different ANNs showed that the visual model's early layer had a higher similarity to the brain in the early stage, while the visual-semantic model's late layer had a higher similarity to the brain in the late stage. In addition, to assess whether this EEG-ANN similarity pattern generalizes to more biologically inspired architectures, we conducted the same analyses and found that CORnet (*Kubilius et al., 2019*) showed similar patterns to those observed for ResNet and CLIP, providing converging evidence across models (*Figure 3—figure supplement 5*). However, for the representation of objects, partial correlation results for different ANNs didn't demonstrate the superiority of the multi-modal model at late layers. This might be due to models like CLIP, which contain semantic information, learning more complex image descriptive information (like the relationship between object and the background in the image). Real-world size might be a semantic dimension of the object itself, and its representation does not require overall semantic descriptive information of the image. In contrast, retinal size and real-world depth could rely on image background information for estimation, thus their representations in the CLIP late layer disappeared when input images had only pure object but no background.

Although our study does not directly test specific models of visual object processing, the observed temporal dynamics provide important constraints for theoretical interpretations. In particular, we find that real-world size representations emerge significantly later than low-level visual features such as retinal size and depth. This temporal profile is difficult to reconcile with a purely feedforward account of visual processing (e.g. *DiCarlo et al., 2012*), which posits that object properties are rapidly computed in a sequential hierarchy of increasingly complex visual features. Instead, our results are more consistent with frameworks that emphasize recurrent or top-down processing, such as the reverse hierarchy theory (*Hochstein and Ahissar, 2002*), which suggests that high-level conceptual information may emerge later and involve feedback to earlier visual areas. This interpretation is further supported by representational similarities with late-stage artificial neural network layers and with a semantic word embedding model (Word2Vec), both of which reflect learned, abstract knowledge rather than low-level visual features. Taken together, these findings suggest that real-world size is not merely a perceptual attribute, but one that draws on conceptual or semantic-level representations acquired through experience. While our EEG analyses focused on posterior electrodes and thus cannot definitively localize cortical sources, we see this study as a step toward linking low-level visual input with higher-level semantic knowledge. Future work incorporating broader spatial coverage (e.g. anterior sensors), source localization, or complementary modalities such as MEG and fMRI will be critical to adjudicate between alternative models of object representation and to more precisely trace the origin and flow of real-world size information in the brain. Moreover, building on the promising findings of our study, future work may further delve into the detailed processes of object processing and object space. One important problem to solve is how real-world size interacts with other object dimensions in object space. In addition, our approach could be used with future studies investigating other influences on object processing, such as how different task conditions impact and modulate the processing of various visual features.

Moreover, we must also emphasize that in this study, we were concerned with perceived real-world size and depth reflecting a perceptual estimation of our world, which are slightly different from absolute physical size and depth. The differences in brain encoding between perceived and absolute physical size and depth require more comprehensive measurements of an object's physical attributes for further exploration. Also, we focused on perceiving depth and size from 2D images in this study, which might have some differences in brain mechanism compared to physically exploring the 3D world. Nevertheless, we believe our study offers a valuable contribution to object recognition, especially the encoding process of object real-world size in natural images. Additionally, we acknowledge that our

metric for real-world depth was derived indirectly as the ratio of perceived real-world size to retinal size. While this formulation is grounded in geometric principles of perspective projection and served the purpose of analytically dissociating depth from size in our RSA framework, it remains a proxy rather than a direct measure of perceived egocentric distance. Future work incorporating behavioral or psychophysical depth ratings would be valuable for validating and refining this metric.

In conclusion, we used computational methods to distinguish the representations of real-world size, retinal size, and real-world depth features of objects in ecologically natural images in both human brains and ANNs. We found an unconfounded representation of object real-world size, which emerged at later time windows in the human EEG signal and at later layers of artificial neural networks compared to real-world depth, and which also appeared to be preserved as a stable dimension in object space. Thus, although size and depth properties are closely correlated, the processing of perceived object size and depth may arise through dissociated time courses and mechanisms. Our research provides a temporally resolved characterization of how certain key object properties – such as object real-world size, depth, and retinal size – are represented in the brain, which advances our understanding of real-world size encoding as a stable and semantically grounded dimension in object space, and contributes to the development of more brain-like visual models.

## Materials and methods

### Experimental design, stimuli images, and EEG data

We utilized the open dataset from THINGS EEG2 (*Gifford et al., 2022*), which includes EEG data from 10 healthy human subjects (age = 28.5 ± 4, 8 female and 2 male) participating in a rapid serial visual presentation (RSVP) paradigm with an orthogonal target detection task to ensure participants paid attention to the visual stimuli. All participants were seated at a fixed distance of 0.6 m from the screen throughout the experiment. For each trial, subjects viewed one image (sized 500×500 pixels) for 100ms. Each subject viewed 16740 images of objects on a natural background for 1854 object concepts from the THINGS dataset (*Hebart et al., 2019*). For the current study, we used the 'test' dataset portion, which includes 16,000 trials per subject corresponding to 200 images (200 object concepts, one image per concept) with 80 trials per image, providing high trial counts per condition necessary for reliable EEG decoding. In contrast, images in the training set were only shown four times each. This design choice ensured higher decoding reliability and greater signal quality for RSA. Before inputting the images to the ANNs, we reshaped image sizes to 224x224 pixels and normalized the pixel values of images to ImageNet statistics.

EEG data were collected using a 64-channel EASYCAP and a BrainVision actiCHamp amplifier. The EEG data were originally sampled at 1000 Hz and online-filtered between 0.1 Hz and 100 Hz during acquisition, with recordings referenced to the Fz electrode. For preprocessing, no additional filtering was applied. Baseline correction was performed by subtracting the mean signal during the 100ms pre-stimulus interval from each trial and channel separately. We already used pre-processed data from 17 channels with labels beginning with 'O' or 'P' (O1, Oz, O2, PO7, PO3, POz, PO4, PO8, P7, P5, P3, P1, Pz, P2) ensuring full coverage of posterior regions typically involved in visual object processing. The epoched data were then down-sampled to 100 Hz. We re-epoched EEG data ranging from 100ms before stimulus onset to 300ms after onset with a sample frequency of 100 Hz. Thus, the shape of our EEG data matrix for each trial was 17 channels ×40 time points.

### ANN models

We applied two pre-trained ANN models: one visual model (ResNet-101 *He et al., 2016* pretrained on ImageNet), and one multi-modal (visual +semantic) model (CLIP with a ResNet-101 backbone (*Radford et al., 2021*) pretrained on YFCC-15M). Our motivation for selecting ResNet-50 and CLIP ResNet-50 was not to make a definitive comparison between model classes, but rather to include two widely used representatives of their respective categories—one trained purely on visual information (ResNet-50 on ImageNet) and one trained with joint visual and linguistic supervision (CLIP ResNet-50 on image–text pairs). These models are both highly influential and commonly used in computational and cognitive neuroscience, allowing for relevant comparisons with existing work (*Choksi et al., 2022a*; *Choksi et al., 2022b*; *Conwell et al., 2024*; *Song et al., 2024*; *Wang et al., 2022b*). We used THINGSvision (*Muttenthaler and Hebart, 2021*) to obtain low- and high-level feature vectors

of ANN activations from early and late layers (early layer: second convolutional layer; late layer: last visual layer) for the images.

## Word2Vec model

To approximate the non-visual, pure semantic space of objects, we also applied a Word2Vec, a natural language processing model for word embedding, pretrained on Google News corpus (*Mikolov et al., 2013*), which contains 300-dimensional vectors for 3 million words and phrases. We input the words for each image's object concept (pre-labeled in THINGS dataset: *Hebart et al., 2019*), instead of the visual images themselves. We used Gensim (*Řehůřek and Sojka, 2010*) to obtain Word2Vec feature vectors for the objects in images.

## Representational dissimilarity matrices (RDMs)

To conduct RSA across human EEG, artificial models, and our hypotheses corresponding to different visual features, we first computed representational dissimilarity matrices (RDMs) for different modalities (*Figure 2*). The shape of each RDM was 200×200, corresponding to pairwise dissimilarity between the 200 images. We extracted the 19,900 cells from the upper half of the diagonal of each RDM for subsequent analyses.

### Neural RDMs

From the EEG signal, we constructed timepoint-by-timepoint neural RDMs for each subject with decoding accuracy as the dissimilarity index (*Figure 2A*). Since EEG has a low SNR and includes rapid transient artifacts, Pearson correlations computed over very short time windows yield unstable dissimilarity estimates (*Kappenman and Luck, 2010*; *Luck, 2014*) and may thus fail to reliably detect differences between images. In contrast, decoding accuracy – by training classifiers to focus on task-relevant features – better mitigates noise and highlights representational differences. We first conducted timepoint-by-timepoint classification-based decoding for each subject and each pair of images (200 images, 19,900 pairs in total). We applied linear Support Vector Machine (SVM) to train and test a two-class classifier, employing a five-time fivefold cross-validation method, to obtain an independent decoding accuracy for each image pair and each timepoint. Therefore, we ultimately acquired 40 (1 per timepoint) EEG RDMs for each subject.

### Hypothesis-based (HYP) RDMs

We constructed three hypothesis-based RDMs reflecting the different types of visual object properties in the naturalistic images (*Figure 2B*): Real-World Size RDM, Retinal Size RDM, and Real-World Depth RDM. We constructed these RDMs as follows:

1. For Real-World Size RDM, we obtained human behavioral real-world size ratings of each object concept from the THINGS +dataset (*Stoinski et al., 2024*). In the THINGS +dataset, 2010 participants (different from the subjects in THINGS EEG2) did an online size rating task and completed a total of 13,024 trials corresponding to 1854 object concepts using a two-step procedure. In their experiment, first, each object was rated on a 520-unit continuous slider anchored by familiar reference objects (e.g. 'grain of sand', 'microwave oven', 'aircraft carrier') representing a logarithmic size range. Participants were not shown numerical values but used semantic anchors as guides. In the second step, the scale zoomed in around the selected region to allow for finer-grained refinement of the size judgment. Final size values were derived from aggregated behavioral data and rescaled to a range of 0–519 for consistency across objects, with the actual mean ratings across subjects ranging from 100.03 ('grain of sand') to 423.09 ('subway'). We used these ratings as the perceived real-world size measure of the object concept pre-labeled in the THINGS dataset (*Hebart et al., 2019*) for each image. We then constructed the representational dissimilarity matrix by calculating the absolute difference between perceived real-world size ratings for each pair of images.

2. For Retinal Size RDM, we applied Adobe Photoshop (Adobe Inc, 2019) to crop objects corresponding to object labels from images manually. All cropping and measurement were conducted by one of the authors to ensure consistency across the dataset. The cropped object images have been made publicly available at https://github.com/ZitongLu1996/RWsize, copy archived at *Lu, 2024* to facilitate future reuse and reproducibility. For each image, we obtained a rectangular region that precisely contains a single object, then measured the diagonal length of the

segmented object in pixels as the retinal size measure (*Konkle and Oliva, 2011*). Due to our calculations being at the object level, if there were more than one same objects in an image, we cropped the most complete one to get more accurate retinal size. We then constructed the RDM by calculating the absolute difference between measured retinal size for each pair of images.

3. For Real-World Depth RDM, we calculated the perceived depth based on the measured retinal size index and behavioral real-world size ratings, such that real-world depth / visual image depth = real-world size / retinal size. Since visual image depth (viewing distance) is held constant across images in the task, inferred real-world depth is proportional to real-world size / retinal size. We then constructed the RDM by calculating the absolute difference between inferred real-world depth index for each pair of images.

## ANN (and Word2Vec) model RDMs

We constructed a total of five model-based RDMs (*Figure 2C*). Our primary analyses used four ANN RDMs, corresponding to the early and late layers for both ResNet and CLIP (*Figure 1—figure supplement 1*). The early layer in ResNet refers to ResNet.maxpool layer, and the late layer in ResNet refers to ResNet.avgpool layer. The early layer in CLIP refers to CLIP.visual.avgpool layer, and the late layer in CLIP refers to CLIP.visual.attnpool layer. We also calculated a single Word2Vec RDM for the pure semantic analysis (*Figure 1—figure supplement 2*). For each RDM, we got the dissimilarities by calculating 1 – Pearson correlation coefficient between each pair of two vectors of the model features corresponding to two input images.

## Representational similarity analyses (RSA) and statistical analyses

We conducted cross-modal representational similarity analyses between the three types of RDMs (*Figure 2*). All decoding and RSA analyses were implemented using NeuroRA (*Lu and Ku, 2020*).

## EEG × ANN (or W2V) RSA

To measure the representational similarity between human brains and ANNs and confirm that ANNs have significantly similar representations to human brains, we calculated the Spearman correlation between the 40 timepoint-by-timepoint EEG neural RDMs and the 4 ANN RDMs corresponding to the representations of ResNet early layer, ResNet late layer, CLIP early layer, CLIP late layer, respectively. We also calculated temporal representational similarity between human brains (EEG RDMs) and the Word2Vec model RDM. Cluster-based permutation tests were conducted to determine the time windows of significant representational similarity. First, we performed one-sample t-tests (one-tailed testing) against zero to get the t-value for each timepoint and extracted significant clusters. We computed the clustering statistic as the sum of t-values in each cluster. Then we conducted 1000 permutations of each subject's timepoint-by-timepoint similarities to calculate a null distribution of maximum clustering statistics. Finally, we assigned cluster-level p-values to each cluster of the actual representational time course by comparing its cluster statistic with the null distribution. Time windows were determined to be significant if the p-value of the corresponding cluster was <0.05.

## EEG × HYP RSA

To evaluate how human brains temporally represent different visual features, we calculated the time course of representational similarity between the timepoint-by-timepoint EEG neural RDMs and the three hypothesis-based RDMs. To avoid correlations between hypothesis-based RDMs (*Figure 3A*) influencing comparison results, we calculated partial correlations and used a one-tailed test against the null hypothesis that the partial correlation was less than or equal to zero, testing whether the partial correlation was significantly greater than zero. In EEG ×HYP partial correlation (*Figure 3D*), we correlated EEG RDMs with one hypothesis-based RDM (e.g. real-world size), while controlling for the other two (e.g. retinal size and real-world depth). Cluster-based permutation tests were performed as described above to determine the time windows of significant representational similarity. In addition, we conducted peak latency analysis to determine the latency of peak representational similarity for each type of visual information with the EEG signal. To assess the stability of peak latency estimates for each subject, we performed a bootstrap procedure across stimulus pairs. At each time point, the EEG RDM was vectorized by extracting the lower triangle (excluding the diagonal), resulting in 19,900

unique pairwise values. For each bootstrap sample, we resampled these 19,900 pairwise entries with replacement to generate a new pseudo-RDM of the same size. We then computed the partial correlation between the EEG pseudo-RDM and a given hypothesis RDM (e.g. real-world size), controlling for other feature RDMs, and obtained a time course of partial correlations. Repeating this procedure 1000 times and extracting the peak latency within the significant time window yielded a distribution of bootstrapped latencies, from which we got the bootstrapped mean latencies per subject. Paired t-tests (two-tailed) were conducted to assess the statistical differences in peak latencies between different visual features.

### ANN (or W2V) × HYP RSA

To evaluate how different visual information is represented in ANNs, we calculated representational similarity between the ANN RDMs and hypothesis-based RDMs. As in the EEG ×HYP RSA, we calculated partial correlations to avoid correlations between hypothesis-based RDMs. In ANN (or W2V)×HYP partial correlation (*Figures 3E and 5A*), we correlated ANN (or W2V) RDMs with one hypothesis-based RDM (e.g. real-world size), while partialling out the other two. We also calculated the partial correlations between hypothesis-based RDMs and the Word2Vec RDM. To determine statistical significance, we conducted a bootstrap test. We shuffled the order of the cells above the diagonal in each ANN (or Word2Vec) RDM 1000 times. For each iteration, we calculated partial correlations corresponding to the three hypothesis-based RDMs. This produced a 1000-sample null distribution for each HYP x ANN (or W2V) RSA. We hypothesized that if the real similarity was higher than the 95% confidence interval of the null distribution, it indicated that ANN (or W2V) features validly encoded the corresponding visual feature.

Additionally, to explore how the naturalistic background present in the images might influence object real-world size, retinal size, and real-world depth representations, we conducted another version of the analysis by inputting cropped object images without background into ANN models to obtain object-only ANN RDMs (*Figure 4—figure supplement 1*). Then we performed the same ANN x HYP similarity analysis to calculate partial correlations between the hypothesis-based RDMs and object-only ANN RDM. (We didn't conduct the similarity analysis between timepoint-by-timepoint EEG neural RDMs with subjects viewing natural images and object-only ANN RDMs due to the input differences.)

## Acknowledgements

This work was supported by research grants from the National Institutes of Health (R01-EY025648) and from the National Science Foundation (NSF 1848939) to JDG. The authors declare no competing financial interests.

## Additional information

### Funding

| Funder | Grant reference number | Author |
| --- | --- | --- |
| National Institutes of Health | R01-EY025648 | Julie Golomb |
| U.S. National Science Foundation | 1848939 | Julie Golomb |

The funders had no role in study design, data collection and interpretation, or the decision to submit the work for publication.

### Author contributions

Zitong Lu, Conceptualization, Formal analysis, Investigation, Visualization, Methodology, Writing – original draft, Project administration, Writing – review and editing; Julie Golomb, Resources, Software, Supervision, Funding acquisition, Project administration, Writing – review and editing

## Author ORCIDs
Zitong Lu ⓘ https://orcid.org/0000-0002-7953-6742
Julie Golomb ⓘ https://orcid.org/0000-0003-3489-0702

Reviewer #1 (Public review): https://doi.org/10.7554/eLife.98117.3.sa1
Reviewer #3 (Public review): https://doi.org/10.7554/eLife.98117.3.sa2
Author response https://doi.org/10.7554/eLife.98117.3.sa3

---

## Additional files

### Supplementary files
MDAR checklist

Supplementary file 1. Statistical results of similarities between ANN and hypothesis-based RDMs.

### Data availability
EEG data and images from THINGS EEG2 data are publicly available on OSF. All Python analysis scripts are publicly available on GitHub (copy archived at *Lu, 2024*).

The following previously published datasets were used:

| Author(s) | Year | Dataset title | Dataset URL | Database and Identifier |
|---|---|---|---|---|
| Hebart MN, Dickter AH, Kidder A, Kwok WY, Corriveau A, Van Wicklin C | 2019 | THINGS object concept and object image database | https://doi.org/10.17605/OSF.IO/JUM2F | Open Science Framework, 10.17605/OSF.IO/JUM2F |
| Gifford AT, Dwivedi K, Roig G, Cichy RM | 2022 | A large and rich EEG dataset for modeling human visual object recognition | https://doi.org/10.17605/OSF.IO/3JK45 | Open Science Framework, 10.17605/OSF.IO/3JK45 |

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
