## [Editor Report · eLife Assessment]

This **important** study combines EEG, neural networks and multivariate pattern analysis to show that real-world size, retinal size and real-world depth are represented at different latencies. The evidence presented is **convincing** and the work will be of broader interest to the experimental and computational vision community.

---

## [Referee Report · Reviewer #1 (Public review)]

Lu & Golomb combined EEG, artificial neural networks, and multivariate pattern analyses to examine how different visual variables are processed in the brain. The conclusions of the paper are mostly well supported.

The authors find that not only real-world size is represented in the brain (which was known), but both retinal size and real-world depth is represented, at different time points or latencies, which may reflect different stages of processing. Prior work has not been able to answer the question of real-world depth due to stimuli used. The authors made this possible by assess real-world depth and testing it with appropriate methodology, accounting for retinal and real-world size. The methodological approach combining behavior, RSA, and ANNs is creative and well thought out to appropriately assess the research questions, and the findings may be very compelling if backed up with some clarifications and further analyses.

The work will be of interest to experimental and computational vision scientists, as well as the broader computational cognitive neuroscience community as the methodology is of interest and the code is or will be made available. The work is important as it is currently not clear what the correspondence between many deep neural network models are and the brain are, and this work pushes our knowledge forward on this front. Furthermore, the availability of methods and data will be useful for the scientific community.

---

## [Referee Report · Reviewer #3 (Public review)]

The authors used an open EEG dataset of observers viewing real-world objects. Each object had a real-world size value (from human rankings), a retinal size value (measured from each image), and a scene depth value (inferred from the above). The authors combined the EEG and object measurements with extant, pre-trained models (a deep convolutional neural network, a multimodal ANN, and Word2vec) to assess the time course of processing object size (retinal and real-world) and depth. They found that depth was processed first, followed by retinal size, and then real-world size. The depth time course roughly corresponded to the visual ANNs, while the real-world size time course roughly corresponded to the more semantic models.

The time course result for the three object attributes is very clear and a novel contribution to the literature. The authors have revised the ANN motivations to increase clarity. Additionally, the authors have appropriately toned down some of the language about novelty, and the addition of a noise ceiling has helped the robustness of the work.

While I appreciate the addition of Cornet in the Supplement, I am less compelled by the authors' argument for Word2Vec over LLMs for "pure" semantic embeddings. While I'm not digging in on this point, this choice may prematurely age this work.

---

## [Author Response]

The following is the authors’ response to the original reviews.

**Reviewer #1 (Public Review):**
Lu & Golomb combined EEG, artificial neural networks, and multivariate pattern analyses to examine how different visual variables are processed in the brain. The conclusions of the paper are mostly well supported, but some aspects of methods and data analysis would benefit from clarification and potential extensions.The authors find that not only real-world size is represented in the brain (which was known), but both retinal size and real-world depth are represented, at different time points or latencies, which may reflect different stages of processing. Prior work has not been able to answer the question of real-world depth due to the stimuli used. The authors made this possible by assessing real-world depth and testing it with appropriate methodology, accounting for retinal and real-world size. The methodological approach combining behavior, RSA, and ANNs is creative and well thought out to appropriately assess the research questions, and the findings may be very compelling if backed up with some clarifications and further analyses.The work will be of interest to experimental and computational vision scientists, as well as the broader computational cognitive neuroscience community as the methodology is of interest and the code is or will be made available. The work is important as it is currently not clear what the correspondence between many deep neural network models and the brain is, and this work pushes our knowledge forward on this front. Furthermore, the availability of methods and data will be useful for the scientific community.
**Reviewer #2 (Public Review):**
Summary:This paper aims to test if neural representations of images of objects in the human brain contain a 'pure' dimension of real-world size that is independent of retinal size or perceived depth. To this end, they apply representational similarity analysis on EEG responses in 10 human subjects to a set of 200 images from a publicly available database (THINGS-EEG2), correlating pairwise distinctions in evoked activity between images with pairwise differences in human ratings of real-world size (from THINGS+). By partialling out correlations with metrics of retinal size and perceived depth from the resulting EEG correlation time courses, the paper claims to identify an independent representation of real-world size starting at 170 ms in the EEG signal. Further comparisons with artificial neural networks and language embeddings lead the authors to claim this correlation reflects a relatively 'high-level' and 'stable' neural representation.Strengths:The paper features insightful figures/illustrations and clear figures.The limitations of prior work motivating the current study are clearly explained and seem reasonable (although the rationale for why using 'ecological' stimuli with backgrounds matters when studying real-world size could be made clearer; one could also argue the opposite, that to get a 'pure' representation of the real-world size of an 'object concept', one should actually show objects in isolation).The partial correlation analysis convincingly demonstrates how correlations between feature spaces can affect their correlations with EEG responses (and how taking into account these correlations can disentangle them better).The RSA analysis and associated statistical methods appear solid.Weaknesses:The claim of methodological novelty is overblown. Comparing image metrics, behavioral measurements, and ANN activations against EEG using RSA is a commonly used approach to study neural object representations. The dataset size (200 test images from THINGS) is not particularly large, and neither is comparing pre-trained DNNs and language models, or using partial correlations.

Thanks for your feedback. We agree that the methods used in our study – such as RSA, partial correlations, and the use of pretrained ANN and language models – are indeed well-established in the literature. We therefore revised the manuscript to more carefully frame our contribution: rather than emphasizing methodological novelty in isolation, we now highlight the combination of techniques, the application to human EEG data with naturalistic images, and the explicit dissociation of real-world size, retinal size, and depth representations as the primary strengths of our approach. Corresponding language in the Abstract, Introduction, and Discussion has been adjusted to reflect this more precise positioning:

(Abstract, line 34 to 37) “our study combines human EEG and representational similarity analysis to disentangle neural representations of object real-world size from retinal size and perceived depth, leveraging recent datasets and modeling approaches to address challenges not fully resolved in previous work.”

(Introduction, line 104 to 106) “we overcome these challenges by combining human EEG recordings, naturalistic stimulus images, artificial neural networks, and computational modeling approaches including representational similarity analysis (RSA) and partial correlation analysis …”

(Introduction, line 108) “We applied our integrated computational approach to an open EEG dataset…”

(Introduction, line 142 to 143) “The integrated computational approach by cross-modal representational comparisons we take with the current study…”

(Discussion, line 550 to 552) “our study goes beyond the contributions of prior studies in several key ways, offering both theoretical and methodological advances: …”

The claims also seem too broad given the fairly small set of RDMs that are used here (3 size metrics, 4 ANN layers, 1 Word2Vec RDM): there are many aspects of object processing not studied here, so it's not correct to say this study provides a 'detailed and clear characterization of the object processing process'.

Thanks for pointing this out. We softened language in our manuscript to reflect that our findings provide a temporally resolved characterization of selected object features, rather than a comprehensive account of object processing:

(line 34 to 37) “our study combines human EEG and representational similarity analysis to disentangle neural representations of object real-world size from retinal size and perceived depth, leveraging recent datasets and modeling approaches to address challenges not fully resolved in previous work.”

(line 46 to 48) “Our research provides a temporally resolved characterization of how certain key object properties – such as object real-world size, depth, and retinal size – are represented in the brain, …”

The paper lacks an analysis demonstrating the validity of the real-world depth measure, which is here computed from the other two metrics by simply dividing them. The rationale and logic of this metric is not clearly explained. Is it intended to reflect the hypothesized egocentric distance to the object in the image if the person had in fact been 'inside' the image? How do we know this is valid? It would be helpful if the authors provided a validation of this metric.

We appreciate the comment regarding the real-world depth metric. Specifically, this metric was computed as the ratio of real-world size (obtained via behavioral ratings) to measured retinal size. The rationale behind this computation is grounded in the basic principles of perspective projection: for two objects subtending the same retinal size, the physically larger object is presumed to be farther away. This ratio thus serves as a proxy for perceived egocentric depth under the simplifying assumption of consistent viewing geometry across images.

We acknowledge that this is a derived estimate and not a direct measurement of perceived depth. While it provides a useful approximation that allows us to analytically dissociate the contributions of real-world size and depth in our RSA framework, we agree that future work would benefit from independent perceptual depth ratings to validate or refine this metric. We added more discussions about this to our revised manuscript:

(line 652 to 657) “Additionally, we acknowledge that our metric for real-world depth was derived indirectly as the ratio of perceived real-world size to retinal size. While this formulation is grounded in geometric principles of perspective projection and served the purpose of analytically dissociating depth from size in our RSA framework, it remains a proxy rather than a direct measure of perceived egocentric distance. Future work incorporating behavioral or psychophysical depth ratings would be valuable for validating and refining this metric.”

Given that there is only 1 image/concept here, the factor of real-world size may be confounded with other things, such as semantic category (e.g. buildings vs. tools). While the comparison of the real-world size metric appears to be effectively disentangled from retinal size and (the author's metric of) depth here, there are still many other object properties that are likely correlated with real-world size and therefore will confound identifying a 'pure' representation of real-world size in EEG. This could be addressed by adding more hypothesis RDMs reflecting different aspects of the images that may correlate with real-world size.

We thank the reviewer for this thoughtful and important point. We agree that semantic category and real-world size may be correlated, and that semantic structure is one of the plausible sources of variance contributing to real-world size representations. However, we would like to clarify that our original goal was to isolate real-world size from two key physical image features — retinal size and inferred real-world depth — which have been major confounds in prior work on this topic. We acknowledge that although our analysis disentangled real-world size from depth and retinal size, this does not imply a fully “pure” representation; therefore, we now refer to the real-world size representations as “partially disentangled” throughout the manuscript to reflect this nuance.

Interestingly, after controlling for these physical features, we still found a robust and statistically isolated representation of real-world size in the EEG signal. This motivated the idea that realworld size may be more than a purely perceptual or image-based property — it may be at least partially semantic. Supporting this interpretation, both the late layers of ANN models and the non-visual semantic model (Word2Vec) also captured real-world size structure. Rather than treating semantic information as an unwanted confound, we propose that semantic structure may be an inherent component of how the brain encodes real-world size.

To directly address the your concern, we conducted an additional variance partitioning analysis, in which we decomposed the variance in EEG RDMs explained by four RDMs: real-world depth, retinal size, real-world size, and semantic information (from Word2Vec). Specifically, for each EEG timepoint, we quantified (1) the unique variance of real-world size, after controlling for semantic similarity, depth, and retinal size; (2) the unique variance of semantic information, after controlling for real-world size, depth, and retinal size; (3) the shared variance jointly explained by real-world size and semantic similarity, controlling for depth and retinal size. This analysis revealed that real-world size explained unique variance in EEG even after accounting for semantic similarity. And there was also a substantial shared variance, indicating partial overlap between semantic structure and size. Semantic information also contributed unique explanatory power, as expected. These results suggest that real-world size is indeed partially semantic in nature, but also has independent neural representation not fully explained by general semantic similarity. This strengthens our conclusion that real-world size functions as a meaningful, higher-level dimension in object representation space.

We now include this new analysis and a corresponding figure (Figure S8) in the revised manuscript:

(line 532 to 539) “Second, we conducted a variance partitioning analysis, in which we decomposed the variance in EEG RDMs explained by three hypothesis-based RDMs and the semantic RDM (Word2Vec RDM), and we still found that real-world size explained unique variance in EEG even after accounting for semantic similarity (Figure S9). And we also observed a substantial shared variance jointly explained by real-world size and semantic similarity and a unique variance of semantic information. These results suggest that real-world size is indeed partially semantic in nature, but also has independent neural representation not fully explained by general semantic similarity.”

The choice of ANNs lacks a clear motivation. Why these two particular networks? Why pick only 2 somewhat arbitrary layers? If the goal is to identify more semantic representations using CLIP, the comparison between CLIP and vision-only ResNet should be done with models trained on the same training datasets (to exclude the effect of training dataset size & quality; cf Wang et al., 2023). This is necessary to substantiate the claims on page 19 which attributed the differences between models in terms of their EEG correlations to one of them being a 'visual model' vs. 'visual-semantic model'.

We argee that the choice and comparison of models should be better contextualized.

First, our motivation for selecting ResNet-50 and CLIP ResNet-50 was not to make a definitive comparison between model classes, but rather to include two widely used representatives of their respective categories—one trained purely on visual information (ResNet-50 on ImageNet) and one trained with joint visual and linguistic supervision (CLIP ResNet-50 on image–text pairs). These models are both highly influential and commonly used in computational and cognitive neuroscience, allowing for relevant comparisons with existing work (line 181-187).

Second, we recognize that limiting the EEG × ANN correlation analyses to only early and late layers may be viewed as insufficiently comprehensive. To address this point, we have computed the EEG correlations with multiple layers in both ResNet and CLIP models (ResNet: ResNet.maxpool, ResNet.layer1, ResNet.layer2, ResNet.layer3, ResNet.layer4, ResNet.avgpool; CLIP: CLIP.visual.avgpool, CLIP.visual.layer1, CLIP.visual.layer2, CLIP.visual.layer3, CLIP.visual.layer4, CLIP.visual.attnpool). The results, now included in Figure S4, show a consistent trend: early layers exhibit higher similarity to early EEG time points, and deeper layers show increased similarity to later EEG stages. We chose to highlight early and late layers in the main text to simplify interpretation.

Third, we appreciate the reviewer’s point that differences in training datasets (ImageNet vs. CLIP's dataset) may confound any attribution of differences in brain alignment to the models' architectural or learning differences. We agree that the comparisons between models trained on matched datasets (e.g., vision-only vs. multimodal models trained on the same image–text corpus) would allow for more rigorous conclusions. Thus, we explicitly acknowledged this limitation in the text:

(line 443 to 445) “However, it is also possible that these differences between ResNet and CLIP reflect differences in training data scale and domain.”

The first part of the claim on page 22 based on Figure 4 'The above results reveal that realworld size emerges with later peak neural latencies and in the later layers of ANNs, regardless of image background information' is not valid since no EEG results for images without backgrounds are shown (only ANNs).

We revised the sentence to clarify that this is a hypothesis based on the ANN results, not an empirical EEG finding:

(line 491 to 495) “These results show that real-world size emerges in the later layers of ANNs regardless of image background information, and – based on our prior EEG results – although we could not test object-only images in the EEG data, we hypothesize that a similar temporal profile would be observed in the brain, even for object-only images.”

While we only had the EEG data of human subjects viewing naturalistic images, the ANN results suggest that real-world size representations may still emerge at later processing stages even in the absence of background, consistent with what we observed in EEG under with-background conditions.

The paper is likely to impact the field by showcasing how using partial correlations in RSA is useful, rather than providing conclusive evidence regarding neural representations of objects and their sizes.Additional context important to consider when interpreting this work:Page 20, the authors point out similarities of peak correlations between models 'Interestingly, the peaks of significant time windows for the EEG × HYP RSA also correspond with the peaks of the EEG × ANN RSA timecourse (Figure 3D,F)'. Although not explicitly stated, this seems to imply that they infer from this that the ANN-EEG correlation might be driven by their representation of the hypothesized feature spaces. However this does not follow: in EEG-image metric model comparisons it is very typical to see multiple peaks, for any type of model, this simply reflects specific time points in EEG at which visual inputs (images) yield distinctive EEG amplitudes (perhaps due to stereotypical waves of neural processing?), but one cannot infer the information being processed is the same. To investigate this, one could for example conduct variance partitioning or commonality analysis to see if there is variance at these specific timepoints that is shared by a specific combination of the hypothesis and ANN feature spaces.

Thanks for your thoughtful observation! Upon reflection, we agree that the sentence – "Interestingly, the peaks of significant time windows for the EEG × HYP RSA also correspond with the peaks of the EEG × ANN RSA timecourse" – was speculative and risked implying a causal link that our data do not warrant. As you rightly points out, observing coincident peak latencies across different models does not necessarily imply shared representational content, given the stereotypical dynamics of evoked EEG responses. And we think even variance partitioning analysis would still not suffice to infer that ANN-EEG correlations are driven specifically by hypothesized feature spaces. Accordingly, we have removed this sentence from the manuscript to avoid overinterpretation.

Page 22 mentions 'The significant time-window (90-300ms) of similarity between Word2Vec RDM and EEG RDMs (Figure 5B) contained the significant time-window of EEG x real-world size representational similarity (Figure 3B)'. This is not particularly meaningful given that the Word2Vec correlation is significant for the entire EEG epoch (from the time-point of the signal 'arriving' in visual cortex around ~90 ms) and is thus much less temporally specific than the realworld size EEG correlation. Again a stronger test of whether Word2Vec indeed captures neural representations of real-world size could be to identify EEG time-points at which there are unique Word2Vec correlations that are not explained by either ResNet or CLIP, and see if those timepoints share variance with the real-world size hypothesized RDM.

We appreciate your insightful comment. Upon reflection, we agree that the sentence – "'The significant time-window (90-300ms) of similarity between Word2Vec RDM and EEG RDMs (Figure 5B) contained the significant time-window of EEG x real-world size representational similarity (Figure 3B)" – was speculative. And we have removed this sentence from the manuscript to avoid overinterpretation.

Additionally, we conducted two analyses as you suggested in the supplement. First, we calculated the partial correlation between EEG RDMs and the Word2Vec RDM while controlling for four ANN RDMs (ResNet early/late and CLIP early/late) (Figure S8). Even after regressing out these ANN-derived features, we observed significant correlations between Word2Vec and EEG RDMs in the 100–190 ms and 250–300 ms time windows. This result suggests that

Word2Vec captures semantic structure in the neural signal that is not accounted for by ResNet or CLIP. Second, we conducted an additional variance partitioning analysis, in which we decomposed the variance in EEG RDMs explained by four RDMs: real-world depth, retinal size, real-world size, and semantic information (from Word2Vec) (Figure S9). And we found significant shared variance between Word2Vec and real-world size at 130–150 ms and 180–250 ms. These results indicate a partially overlapping representational structure between semantic content and real-world size in the brain.

We also added these in our revised manuscript:

(line 525 to 539) “To further probe the relationship between real-world size and semantic information, and to examine whether Word2Vec captures variances in EEG signals beyond that explained by visual models, we conducted two additional analyses. First, we performed a partial correlation between EEG RDMs and the Word2Vec RDM, while regressing out four ANN RDMs (early and late layers of both ResNet and CLIP) (Figure S8). We found that semantic similarity remained significantly correlated with EEG signals across sustained time windows (100-190ms and 250-300ms), indicating that Word2Vec captures neural variance not fully explained by visual or visual-language models. Second, we conducted a variance partitioning analysis, in which we decomposed the variance in EEG RDMs explained by three hypothesis-based RDMs and the semantic RDM (Word2Vec RDM), and we still found that real-world size explained unique variance in EEG even after accounting for semantic similarity (Figure S9). And we also observed a substantial shared variance jointly explained by realworld size and semantic similarity and a unique variance of semantic information. These results suggest that real-world size is indeed partially semantic in nature, but also has independent neural representation not fully explained by general semantic similarity.”

**Reviewer #3 (Public Review):**
The authors used an open EEG dataset of observers viewing real-world objects. Each object had a real-world size value (from human rankings), a retinal size value (measured from each image), and a scene depth value (inferred from the above). The authors combined the EEG and object measurements with extant, pre-trained models (a deep convolutional neural network, a multimodal ANN, and Word2vec) to assess the time course of processing object size (retinal and real-world) and depth. They found that depth was processed first, followed by retinal size, and then real-world size. The depth time course roughly corresponded to the visual ANNs, while the real-world size time course roughly corresponded to the more semantic models.The time course result for the three object attributes is very clear and a novel contribution to the literature. However, the motivations for the ANNs could be better developed, the manuscript could better link to existing theories and literature, and the ANN analysis could be modernized. I have some suggestions for improving specific methods.(1) Manuscript motivationsThe authors motivate the paper in several places by asking " whether biological and artificial systems represent object real-world size". This seems odd for a couple of reasons. Firstly, the brain must represent real-world size somehow, given that we can reason about this question. Second, given the large behavioral and fMRI literature on the topic, combined with the growing ANN literature, this seems like a foregone conclusion and undermines the novelty of this contribution.

Thanks for your helpful comment. We agree that asking whether the brain represents real-world size is not a novel question, given the existing behavioral and neuroimaging evidence supporting this. Our intended focus was not on the existence of real-world size representations per se, but the nature of these representations, particularly the relationship between the temporal dynamics and potential mechanisms of representations of real-world size versus other related perceptual properties (e.g., retinal size and real-world depth). We revised the relevant sentence to better reflect our focue, shifting from a binary framing (“whether or not size is represented”) to a more mechanistic and time-resolved inquiry (“how and when such representations emerge”):

(line 144 to 149) “Unraveling the internal representations of object size and depth features in both human brains and ANNs enables us to investigate how distinct spatial properties—retinal size, realworld depth, and real-world size—are encoded across systems, and to uncover the representational mechanisms and temporal dynamics through which real-world size emerges as a potentially higherlevel, semantically grounded feature.”

While the introduction further promises to "also investigate possible mechanisms of object realworld size representations.", I was left wishing for more in this department. The authors report correlations between neural activity and object attributes, as well as between neural activity and ANNs. It would be nice to link the results to theories of object processing (e.g., a feedforward sweep, such as DiCarlo and colleagues have suggested, versus a reverse hierarchy, such as suggested by Hochstein, among others). What is semantic about real-world size, and where might this information come from? (Although you may have to expand beyond the posterior electrodes to do this analysis).

We thank the reviewer for this insightful comment. We agree that understanding the mechanisms underlying real-world size representations is a critical question. While our current study does not directly test specific theoretical frameworks such as the feedforward sweep model or the reverse hierarchy theory, our results do offer several relevant insights: The temporal dynamics revealed by EEG—where real-world size emerges later than retinal size and depth—suggest that such representations likely arise beyond early visual feedforward stages, potentially involving higherlevel semantic processing. This interpretation is further supported by the fact that real-world size is strongly captured by late layers of ANNs and by a purely semantic model (Word2Vec), suggesting its dependence on learned conceptual knowledge.

While we acknowledge that our analyses were limited to posterior electrodes and thus cannot directly localize the cortical sources of these effects, we view this work as a first step toward bridging low-level perceptual features and higher-level semantic representations. We hope future work combining broader spatial sampling (e.g., anterior EEG sensors or source localization) and multimodal recordings (e.g., MEG, fMRI) can build on these findings to directly test competing models of object processing and representation hierarchy.

We also added these to the Discussion section:

(line 619 to 638) “Although our study does not directly test specific models of visual object processing, the observed temporal dynamics provide important constraints for theoretical interpretations. In particular, we find that real-world size representations emerge significantly later than low-level visual features such as retinal size and depth. This temporal profile is difficult to reconcile with a purely feedforward account of visual processing (e.g., DiCarlo et al., 2012), which posits that object properties are rapidly computed in a sequential hierarchy of increasingly complex visual features. Instead, our results are more consistent with frameworks that emphasize recurrent or top-down processing, such as the reverse hierarchy theory (Hochstein & Ahissar, 2002), which suggests that high-level conceptual information may emerge later and involve feedback to earlier visual areas. This interpretation is further supported by representational similarities with late-stage artificial neural network layers and with a semantic word embedding model (Word2Vec), both of which reflect learned, abstract knowledge rather than low-level visual features. Taken together, these findings suggest that real-world size is not merely a perceptual attribute, but one that draws on conceptual or semantic-level representations acquired through experience. While our EEG analyses focused on posterior electrodes and thus cannot definitively localize cortical sources, we see this study as a step toward linking low-level visual input with higher-level semantic knowledge. Future work incorporating broader spatial coverage (e.g., anterior sensors), source localization, or complementary modalities such as MEG and fMRI will be critical to adjudicate between alternative models of object representation and to more precisely trace the origin and flow of real-world size information in the brain.”

Finally, several places in the manuscript tout the "novel computational approach". This seems odd because the computational framework and pipeline have been the most common approach in cognitive computational neuroscience in the past 5-10 years.

We have revised relevant statements throughout the manuscript to avoid overstating novelty and to better reflect the contribution of our study.

(2) Suggestion: modernize the approachI was surprised that the computational models used in this manuscript were all 8-10 years old. Specifically, because there are now deep nets that more explicitly model the human brain (e.g., Cornet) as well as more sophisticated models of semantics (e.g., LLMs), I was left hoping that the authors had used more state-of-the-art models in the work. Moreover, the use of a single dCNN, a single multi-modal model, and a single word embedding model makes it difficult to generalize about visual, multimodal, and semantic features in general.

Thanks for your suggestion. Indeed, our choice of ResNet and CLIP was motivated by their widespread use in the cognitive and computational neuroscience area. These models have served as standard benchmarks in many studies exploring correspondence between ANNs and human brain activity. To address you concern, we have now added additional results from the more biologically inspired model, CORnet, in the supplementary (Figure S10). The results for CORnet show similar patterns to those observed for ResNet and CLIP, providing converging evidence across models.

Regarding semantic modeling, we intentionally chose Word2Vec rather than large language models (LLMs), because our goal was to examine concept-level, context-free semantic representations. Word2Vec remains the most widely adopted approach for obtaining noncontextualized embeddings that reflect core conceptual similarity, as opposed to the contextdependent embeddings produced by LLMs, which are less directly suited for capturing stable concept-level structure across stimuli.

(3) Methodological considerations(a) Validity of the real-world size measurementI was concerned about a few aspects of the real-world size rankings. First, I am trying to understand why the scale goes from 100-519. This seems very arbitrary; please clarify. Second, are we to assume that this scale is linear? Is this appropriate when real-world object size is best expressed on a log scale? Third, the authors provide "sand" as an example of the smallest realworld object. This is tricky because sand is more "stuff" than "thing", so I imagine it leaves observers wondering whether the experimenter intends a grain of sand or a sandy scene region. What is the variability in real-world size ratings? Might the variability also provide additional insights in this experiment?

We now clarify the origin, scaling, and interpretation of the real-world size values obtained from the THINGS+ dataset.

In their experiment, participants first rated the size of a single object concept (word shown on the screen) by clicking on a continuous slider of 520 units, which was anchored by nine familiar real-world reference objects (e.g., “grain of sand,” “microwave oven,” “aircraft carrier”) that spanned the full expected size range on a logarithmic scale. Importantly, participants were not shown any numerical values on the scale—they were guided purely by the semantic meaning and relative size of the anchor objects. After the initial response, the scale zoomed in around the selected region (covering 160 units of the 520-point scale) and presented finer anchor points between the previous reference objects. Participants then refined their rating by dragging from the lower to upper end of the typical size range for that object. If the object was standardized in size (e.g., “soccer ball”), a single click sufficed. These size judgments were collected across at least 50 participants per object, and final scores were derived from the central tendency of these responses. Although the final size values numerically range from 0 to 519 (after scaling), this range is not known to participants and is only applied post hoc to construct the size RDMs.

Regarding the term “sand”: the THINGS+ dataset distinguished between object meanings when ambiguity was present. For “sand,” participants were instructed to treat it as “a grain of sand”— consistent with the intended meaning of a discrete, minimal-size reference object.

Finally, we acknowledge that real-world size ratings may carry some degree of variability across individuals. However, the dataset includes ratings from 2010 participants across 1854 object concepts, with each object receiving at least 50 independent ratings. Given this large and diverse sample, the mean size estimates are expected to be stable and robust across subjects. While we did not include variability metrics in our main analysis, we believe the aggregated ratings provide a reliable estimate of perceived real-world size.

We added these details in the Materials and Method section:

(line 219 to 230) “In the THINGS+ dataset, 2010 participants (different from the subjects in THINGS EEG2) did an online size rating task and completed a total of 13024 trials corresponding to 1854 object concepts using a two-step procedure. In their experiment, first, each object was rated on a 520unit continuous slider anchored by familiar reference objects (e.g., “grain of sand,” “microwave oven,” “aircraft carrier”) representing a logarithmic size range. Participants were not shown numerical values but used semantic anchors as guides. In the second step, the scale zoomed in around the selected region to allow for finer-grained refinement of the size judgment. Final size values were derived from aggregated behavioral data and rescaled to a range of 0–519 for consistency across objects, with the actual mean ratings across subjects ranging from 100.03 (‘grain of sand’) to 423.09 (‘subway’).”

(b) This work has no noise ceiling to establish how strong the model fits are, relative to the intrinsic noise of the data. I strongly suggest that these are included.

We have now computed noise ceiling estimates for the EEG RDMs across time. The noise ceiling was calculated by correlating each participant’s EEG RDM with the average EEG RDM across the remaining participants (leave-one-subject-out), at each time point. This provides an upper-bound estimate of the explainable variance, reflecting the maximum similarity that any model—no matter how complex—could potentially achieve, given the intrinsic variability in the EEG data.

Importantly, the observed EEG–model similarity values are substantially below this upper bound. This outcome is fully expected: Each of our model RDMs (e.g., real-world size, ANN layers) captures only a specific aspect of the neural representational structure, rather than attempting to account for the totality of the EEG signal. Our goal is not to optimize model performance or maximize fit, but to probe which components of object information are reflected in the spatiotemporal dynamics of the brain’s responses.

For clarity and accessibility of the main findings, we present the noise ceiling time courses separately in the supplementary materials (Figure S7). Including them directly in the EEG × HYP or EEG × ANN plots would conflate distinct interpretive goals: the model RDMs are hypothesis-driven probes of specific representational content, whereas the noise ceiling offers a normative upper bound for total explainable variance. Keeping these separate ensures each visualization remains focused and interpretable.

**Reviewer #1 (Recommendations For The Authors)::**
Some analyses are incomplete, which would be improved if the authors showed analyses with other layers of the networks and various additional partial correlation analyses.Clarity(1) Partial correlations methods incomplete - it is not clear what is being partialled out in each analysis. It is possible to guess sometimes, but it is not entirely clear for each analysis. This is important as it is difficult to assess if the partial correlations are sensible/correct in each case. Also, the Figure 1 caption is short and unclear.For example, ANN-EEG partial correlations - "Finally, we directly compared the timepoint-bytimepoint EEG neural RDMs and the ANN RDMs (Figure 3F). The early layer representations of both ResNet and CLIP were significantly correlated with early representations in the human brain" What is being partialled out? Figure 3F says partial correlation

We apologize for the confusion. We made several key clarifications and corrections in the revised version.

First, we identified and corrected a labeling error in both Figure 1 and Figure 3F. Specifically, our EEG × ANN analysis used Spearman correlation, not partial correlation as mistakenly indicated in the original figure label and text. We conducted parital correlations for EEG × HYP and ANN × HYP. But for EEG × ANN, we directly calculated the correlation between EEG RDMs and ANN RDM corresponding to different layers respectively. We corrected these errors: (1) In Figure 1, we removed the erroneous “partial” label from the EEG × ANN path and updated the caption to clearly outline which comparisons used partial correlation. (2) In Figure 3F, we corrected the Y-axis label to “(correlation)”.

Second, to improve clarity, we have now revised the Materials and Methods section to explicitly describe what is partialled out in each parital correlation analysis:

(line 284 to 286) “In EEG × HYP partial correlation (Figure 3D), we correlated EEG RDMs with one hypothesis-based RDM (e.g., real-world size), while controlling for the other two (retinal size and real-world depth).”

(line 303 to 305) “In ANN (or W2V) × HYP partial correlation (Figure 3E and Figure 5A), we correlated ANN (or W2V) RDMs with one hypothesis-based RDM (e.g., real-world size), while partialling out the other two.”

Finally, the caption of Figure 1 has been expanded to clarify the full analysis pipeline and explicitly specify the partial correlation or correlation in each comparison.

(line 327 to 332) “Figure 1 Overview of our analysis pipeline including constructing three types of RDMs and conducting comparisons between them. We computed RDMs from three sources: neural data (EEG), hypothesized object features (real-world size, retinal size, and real-world depth), and artificial models (ResNet, CLIP, and Word2Vec). Then we conducted cross-modal representational similarity analyses between: EEG × HYP (partial correlation, controlling for other two HYP features), ANN (or W2V) × HYP (partial correlation, controlling for other two HYP features), and EEG × ANN (correlation).”

We believe these revisions now make all analytic comparisons and correlation types full clear and interpretable.

Issues / open questions(2) Semantic representations vs hypothesized (hyp) RDMs (real-world size, etc) - are the representations explained by variables in hyp RDMs or are there semantic representations over and above these? E.g., For ANN correlation with the brain, you could partial out hyp RDMs - and assess whether there is still semantic information left over, or is the variance explained by the hyp RDMs?

Thank for this suggestion. As you suggested, we conducted the partial correlation analysis between EEG RDMs and ANN RDMs, controlling for the three hypothesis-based RDMs. The results (Figure S6) revealed that the EEG×ANN representational similarity remained largely unchanged, indicating that ANN representations capture much more additional representational structure not accounted for by the current hypothesized features. This is also consistent with the observation that EEG×HYP partial correlations were themselves small, but EEG×ANN correlations were much greater.

We also added this statement to the main text:

(line 446 to 451) “To contextualize how much of the shared variance between EEG and ANN representations is driven by the specific visual object features we tested above, we conducted a partial correlation analysis between EEG RDMs and ANN RDMs controlling for the three hypothesis-based RDMs (Figure S6). The EEG×ANN similarity results remained largely unchanged, suggesting that ANN representations capture much more additional rich representational structure beyond these features. ”

(3) Why only early and late layers? I can see how it's clearer to present the EEG results. However, the many layers in these networks are an opportunity - we can see how simple/complex linear/non-linear the transformation is over layers in these models. It would be very interesting and informative to see if the correlations do in fact linearly increase from early to later layers, or if the story is a bit more complex. If not in the main text, then at least in the supplement.

Thank you for the thoughtful suggestion. To address this point, we have computed the EEG correlations with multiple layers in both ResNet and CLIP models (ResNet: ResNet.maxpool, ResNet.layer1, ResNet.layer2, ResNet.layer3, ResNet.layer4, ResNet.avgpool; CLIP:CLIP.visual.avgpool, CLIP.visual.layer1, CLIP.visual.layer2, CLIP.visual.layer3, CLIP.visual.layer4, CLIP.visual.attnpool). The results, now included in Figure S4 and S5, show a consistent trend: early layers exhibit higher similarity to early EEG time points, and deeper layers show increased similarity to later EEG stages. We chose to highlight early and late layers in the main text to simplify interpretation, but now provide the full layerwise profile for completeness.

(4) Peak latency analysis - Estimating peaks per ppt is presumably noisy, so it seems important to show how reliable this is. One option is to find the bootstrapped mean latencies per subject.

Thanks for your suggestion. To estimate the robustness of peak latency values, we implemented a bootstrap procedure by resampling the pairwise entries of the EEG RDM with replacement. For each bootstrap sample, we computed a new EEG RDM and recalculated the partial correlation time course with the hypothesis RDMs. We then extracted the peak latency within the predefined significant time window. Repeating this process 1000 times allowed us to get the bootstrapped mean latencies per subject as the more stable peak latency result. Notably, the bootstrapped results showed minimal deviation from the original latency estimates, confirming the robustness of our findings. Accordingly, we updated the Figure 3D and added these in the Materials and Methods section:

(line 289 to 298) “To assess the stability of peak latency estimates for each subject, we performed a bootstrap procedure across stimulus pairs. At each time point, the EEG RDM was vectorized by extracting the lower triangle (excluding the diagonal), resulting in 19,900 unique pairwise values. For each bootstrap sample, we resampled these 19,900 pairwise entries with replacement to generate a new pseudo-RDM of the same size. We then computed the partial correlation between the EEG pseudo-RDM and a given hypothesis RDM (e.g., real-world size), controlling for other feature RDMs, and obtained a time course of partial correlations. Repeating this procedure 1000 times and extracting the peak latency within the significant time window yielded a distribution of bootstrapped latencies, from which we got the bootstrapped mean latencies per subject.”

(5) "Due to our calculations being at the object level, if there were more than one of the same objects in an image, we cropped the most complete one to get a more accurate retinal size. " Did EEG experimenters make sure everyone sat the same distance from the screen? and remain the same distance? This would also affect real-world depth measures.

Yes, the EEG dataset we used (THINGS EEG2; Gifford et al., 2022) was collected under carefully controlled experimental conditions. We have confirmed that all participants were seated at a fixed distance of 0.6 meters from the screen throughout the experiment. We also added this information in the method (line 156 to 157).

Minor issues/questions - note that these are not raised in the Public Review(6) Title - less about rigor/quality of the work but I feel like the title could be improved/extended. The work tells us not only about real object size, but also retinal size and depth. In fact, isn't the most novel part of this the real-world depth aspect? Furthermore, it feels like the current title restricts its relevance and impact... Also doesn't touch on the temporal aspect, or processing stages, which is also very interesting. There may be something better, but simply adding something like"...disentangled features of real-world size, depth, and retinal size over time OR processing stages".

Thanks for your suggestion! We changed our title – “Human EEG and artificial neural networks reveal disentangled representations and processing timelines of object real-world size and depth in natural images”.

(7) "Each subject viewed 16740 images of objects on a natural background for 1854 object concepts from the THINGS dataset (Hebart et al., 2019). For the current study, we used the 'test' dataset portion, which includes 16000 trials per subject corresponding to 200 images." Why test images? Worth explaining.

We chose to use the “test set” of the THINGS EEG2 dataset for the following two reasons:

(1) Higher trial count per condition: In the test set, each of the 200 object images was presented 80 times per subject, whereas in the training set, each image was shown only 4 times. This much higher trial count per condition in the test set allows for substantially higher signal-tonoise ratio in the EEG data.

(2) Improved decoding reliability: Our analysis relies on constructing EEG RDMs based on pairwise decoding accuracy using linear SVM classifiers. Reliable decoding estimates require a sufficient number of trials per condition. The test set design is thus better suited to support high-fidelity decoding and robust representational similarity analysis.

We also added these explainations to our revised manuscript (line 161 to 164).

(8) "For Real-World Size RDM, we obtained human behavioral real-world size ratings of each object concept from the THINGS+ dataset (Stoinski et al., 2022).... The range of possible size ratings was from 0 to 519 in their online size rating task..." How were the ratings made? What is this scale - do people know the numbers? Was it on a continuous slider?

We should clarify how the real-world size values were obtained from the THINGS+ dataset.

In their experiment, participants first rated the size of a single object concept (word shown on the screen) by clicking on a continuous slider of 520 units, which was anchored by nine familiar real-world reference objects (e.g., “grain of sand,” “microwave oven,” “aircraft carrier”) that spanned the full expected size range on a logarithmic scale. Importantly, participants were not shown any numerical values on the scale—they were guided purely by the semantic meaning and relative size of the anchor objects. After the initial response, the scale zoomed in around the selected region (covering 160 units of the 520-point scale) and presented finer anchor points between the previous reference objects. Participants then refined their rating by dragging from the lower to upper end of the typical size range for that object. If the object was standardized in size (e.g., “soccer ball”), a single click sufficed. These size judgments were collected across at least 50 participants per object, and final scores were derived from the central tendency of these responses. Although the final size values numerically range from 0 to 519 (after scaling), this range is not known to participants and is only applied post hoc to construct the size RDMs.

We added these details in the Materials and Method section:

(line 219 to 230) “In the THINGS+ dataset, 2010 participants (different from the subjects in THINGS EEG2) did an online size rating task and completed a total of 13024 trials corresponding to 1854 object concepts using a two-step procedure. In their experiment, first, each object was rated on a 520unit continuous slider anchored by familiar reference objects (e.g., “grain of sand,” “microwave oven,” “aircraft carrier”) representing a logarithmic size range. Participants were not shown numerical values but used semantic anchors as guides. In the second step, the scale zoomed in around the selected region to allow for finer-grained refinement of the size judgment. Final size values were derived from aggregated behavioral data and rescaled to a range of 0–519 for consistency across objects, with the actual mean ratings across subjects ranging from 100.03 (‘grain of sand’) to 423.09 (‘subway’).”

(9) "For Retinal Size RDM, we applied Adobe Photoshop (Adobe Inc, 2019) to crop objects corresponding to object labels from images manually... " Was this by one person? Worth noting, and worth sharing these values per image if not already for other researchers as it could be a valuable resource (and increase citations).

Yes, all object cropping were performed consistently by one of the authors to ensure uniformity across images. We agree that this dataset could be a useful resource to the community. We have now made the cropped object images publicly available https://github.com/ZitongLu1996/RWsize.

We also updated the manuscript accordingly to note this (line 236 to 239).

(10) "Neural RDMs. From the EEG signal, we constructed timepoint-by-timepoint neural RDMs for each subject with decoding accuracy as the dissimilarity index " Decoding accuracy is presumably a similarity index. Maybe 1-accuracy (proportion correct) for dissimilarity?

Decoding accuracy is a dissimilarity index instead of a similarity index, as higher decoding accuracy between two conditions indicates that they are more distinguishable – i.e., less similar – in the neural response space. This approach aligns with prior work using classification-based representational dissimilarity measures (Grootswagers et al., 2017; Xie et al., 2020), where better decoding implies greater dissimilarity between conditions. Therefore, there is no need to invert the decoding accuracy values (e.g., using 1 - accuracy).

Grootswagers, T., Wardle, S. G., & Carlson, T. A. (2017). Decoding dynamic brain patterns from evoked responses: A tutorial on multivariate pattern analysis applied to time series neuroimaging data. Journal of Cognitive Neuroscience, 29(4), 677-697.

Xie, S., Kaiser, D., & Cichy, R. M. (2020). Visual imagery and perception share neural representations in the alpha frequency band. Current Biology, 30(13), 2621-2627.

(11) Figure 1 caption is very short - Could do with a more complete caption. Unclear what the partial correlations are (what is being partialled out in each case), what are the comparisons "between them" - both in the figure and the caption. Details should at least be in the main text.

Related to your comment (1). We revised the caption and the corresponding text.

**Reviewer #2 (Recommendations For The Authors):**
(1) Intro:Quek et al., (2023) is referred to as a behavioral study, but it has EEG analyses.

We corrected this – “…, one recent study (Quek et al., 2023) …”

The phrase 'high temporal resolution EEG' is a bit strange - isn't all EEG high temporal resolution? Especially when down-sampling to 100 Hz (40 time points/epoch) this does not qualify as particularly high-res.

We removed this phrasing in our manuscript.

(2) Methods:It would be good to provide more details on the EEG preprocessing. Were the data low-pass filtered, for example?

We added more details to the manuscript:

(line 167 to 174) “The EEG data were originally sampled at 1000Hz and online-filtered between 0.1 Hz and 100 Hz during acquisition, with recordings referenced to the Fz electrode. For preprocessing, no additional filtering was applied. Baseline correction was performed by subtracting the mean signal during the 100 ms pre-stimulus interval from each trial and channel separately. We used already preprocessed data from 17 channels with labels beginning with “O” or “P” (O1, Oz, O2, PO7, PO3, POz, PO4, PO8, P7, P5, P3, P1, Pz, P2) ensuring full coverage of posterior regions typically involved in visual object processing. The epoched data were then down-sampled to 100 Hz.”

It is important to provide more motivation about the specific ANN layers chosen. Were these layers cherry-picked, or did they truly represent a gradual shift over the course of layers?

We appreciate the reviewer’s concern and fully agree that it is important to ensure transparency in how ANN layers were selected. The early and late layers reported in the main text were not cherry-picked to maximize effects, but rather intended to serve as illustrative examples representing the lower and higher ends of the network hierarchy. To address this point directly, we have computed the EEG correlations with multiple layers in both ResNet and CLIP models (ResNet: ResNet.maxpool, ResNet.layer1, ResNet.layer2, ResNet.layer3, ResNet.layer4, ResNet.avgpool; CLIP: CLIP.visual.avgpool, CLIP.visual.layer1, CLIP.visual.layer2, CLIP.visual.layer3, CLIP.visual.layer4, CLIP.visual.attnpool). The results, now included in Figure S4, show a consistent trend: early layers exhibit higher similarity to early EEG time points, and deeper layers show increased similarity to later EEG stages.

It is important to provide more specific information about the specific ANN layers chosen. 'Second convolutional layer': is this block 2, the ReLu layer, the maxpool layer? What is the 'last visual layer'?

Apologize for the confusing! We added more details about the layer chosen:

(line 255 to 257) “The early layer in ResNet refers to ResNet.maxpool layer, and the late layer in ResNet refers to ResNet.avgpool layer. The early layer in CLIP refers to CLIP.visual.avgpool layer, and the late layer in CLIP refers to CLIP.visual.attnpool layer.”

Again the claim 'novel' is a bit overblown here since the real-world size ratings were also already collected as part of THINGS+, so all data used here is available.

We removed this phrasing in our manuscript.

Real-world size ratings ranged 'from 0 - 519'; it seems unlikely this was the actual scale presented to subjects, I assume it was some sort of slider?

You are correct. We should clarify how the real-world size values were obtained from the THINGS+ dataset.

In their experiment, participants first rated the size of a single object concept (word shown on the screen) by clicking on a continuous slider of 520 units, which was anchored by nine familiar real-world reference objects (e.g., “grain of sand,” “microwave oven,” “aircraft carrier”) that spanned the full expected size range on a logarithmic scale. Importantly, participants were not shown any numerical values on the scale—they were guided purely by the semantic meaning and relative size of the anchor objects. After the initial response, the scale zoomed in around the selected region (covering 160 units of the 520-point scale) and presented finer anchor points between the previous reference objects. Participants then refined their rating by dragging from the lower to upper end of the typical size range for that object. If the object was standardized in size (e.g., “soccer ball”), a single click sufficed. These size judgments were collected across at least 50 participants per object, and final scores were derived from the central tendency of these responses. Although the final size values numerically range from 0 to 519 (after scaling), this range is not known to participants and is only applied post hoc to construct the size RDMs.

We added these details in the Materials and Method section:

(line 219 to 230) “In the THINGS+ dataset, 2010 participants (different from the subjects in THINGS EEG2) did an online size rating task and completed a total of 13024 trials corresponding to 1854 object concepts using a two-step procedure. In their experiment, first, each object was rated on a 520unit continuous slider anchored by familiar reference objects (e.g., “grain of sand,” “microwave oven,” “aircraft carrier”) representing a logarithmic size range. Participants were not shown numerical values but used semantic anchors as guides. In the second step, the scale zoomed in around the selected region to allow for finer-grained refinement of the size judgment. Final size values were derived from aggregated behavioral data and rescaled to a range of 0–519 for consistency across objects, with the actual mean ratings across subjects ranging from 100.03 (‘grain of sand’) to 423.09 (‘subway’).”

Why is conducting a one-tailed (p<0.05) test valid for EEG-ANN comparisons? Shouldn't this be two-tailed?

Our use of one-tailed tests was based on the directional hypothesis that representational similarity between EEG and ANN RDMs would be positive, as supported by prior literature showing correspondence between hierarchical neural networks and human brain representations (e.g., Cichy et al., 2016; Kuzovkin et al., 2014). This is consistent with a large number of RSA studies which conduct one-tailed tests (i.e., testing the hypothesis that coefficients were greater than zero: e.g., Kuzovkin et al., 2018; Nili et al., 2014; Hebart et al., 2018; Kaiser et al., 2019; Kaiser et al., 2020; Kaiser et al., 2022). Thus, we specifically tested whether the similarity was significantly greater than zero.

Cichy, R. M., Khosla, A., Pantazis, D., Torralba, A., & Oliva, A. (2016). Comparison of deep neural networks to spatio-temporal cortical dynamics of human visual object recognition reveals hierarchical correspondence. Scientific reports, 6(1), 27755.

Kuzovkin, I., Vicente, R., Petton, M., Lachaux, J. P., Baciu, M., Kahane, P., ... & Aru, J. (2018). Activations of deep convolutional neural networks are aligned with gamma band activity of human visual cortex. Communications biology, 1(1), 107.

Nili, H., Wingfield, C., Walther, A., Su, L., Marslen-Wilson, W., & Kriegeskorte, N. (2014). A toolbox for representational similarity analysis. PLoS computational biology, 10(4), e1003553.

Hebart, M. N., Bankson, B. B., Harel, A., Baker, C. I., & Cichy, R. M. (2018). The representational dynamics of task and object processing in humans. Elife, 7, e32816.

Kaiser, D., Turini, J., & Cichy, R. M. (2019). A neural mechanism for contextualizing fragmented inputs during naturalistic vision. elife, 8, e48182.

Kaiser, D., Inciuraite, G., & Cichy, R. M. (2020). Rapid contextualization of fragmented scene information in the human visual system. Neuroimage, 219, 117045.

Kaiser, D., Jacobs, A. M., & Cichy, R. M. (2022). Modelling brain representations of abstract concepts. PLoS Computational Biology, 18(2), e1009837.

Importantly, we note that using a two-tailed test instead would not change the significance of our results. However, we believe the one-tailed test remains more appropriate given our theoretical prediction of positive similarity between ANN and brain representations.

The sentence on the partial correlation description (page 11 'we calculated partial correlations with one-tailed test against the alternative hypothesis that the partial correlation was positive (greater than zero)') didn't make sense to me; are you referring to the null hypothesis here?

We revised this sentence to clarify that we tested against the null hypothesis that the partial correlation was less than or equal to zero, using a one-tailed test to assess whether the correlation was significantly greater than zero.

(line 281 to 284) “…, we calculated partial correlations and used a one-tailed test against the null hypothesis that the partial correlation was less than or equal to zero, testing whether the partial correlation was significantly greater than zero.”

(3) Results:I would prevent the use of the word 'pure', your measurement is one specific operationalization of this concept of real-world size that is not guaranteed to result in unconfounded representations. This is in fact impossible whenever one is using a finite set of natural stimuli and calculating metrics on those - there can always be a factor or metric that was not considered that could explain some of the variance in your measurement. It is overconfident to claim to have achieved some form of Platonic ideal here and to have taken into account all confounds.

Your point is well taken. Our original use of the term “pure” was intended to reflect statistical control for known confounding factors, but we recognize that this wording may imply a stronger claim than warranted. In response, we revised all relevant language in the manuscript to instead describe the statistically isolated or relatively unconfounded representation of real-world size, clarifying that our findings pertain to the unique contribution of real-world size after accounting for retinal size and real-world depth.

Figure 2C: It's not clear why peak latencies are computed on the 'full' correlations rather than the partial ones.

No. The peak latency results in Figure 2C were computed on the partial correlation results – we mentioned this in the figure caption – “Temporal latencies for peak similarity (partial Spearman correlations) between EEG and the 3 types of object information.”

SEM = SEM across the 10 subjects?

Yes. We added this in the figure caption.

Figure 3F y-axis says it's partial correlations but not clear what is partialled out here.

We identified and corrected a labeling error in both Figure 1 and Figure 3F. Specifically, our EEG × ANN analysis used Spearman correlation, not partial correlation as mistakenly indicated in the original figure label and text. We conducted parital correlations for EEG × HYP and ANN × HYP. But for EEG × ANN, we directly calculated the correlation between EEG RDMs and ANN RDM corresponding to different layers respectively. We corrected these errors: (1) In Figure 1, we removed the erroneous “partial” label from the EEG × ANN path and updated the caption to clearly outline which comparisons used partial correlation. (2) In Figure 3F, we corrected the Y-axis label to “(correlation)”.

**Reviewer #3 (Recommendations For The Authors):**
(1) Several methodologies should be clarified:(a) It's stated that EEG was sampled at 100 Hz. I assume this was downsampled? From what original frequency?

Yes. We added more detailed about EEG data:

(line 167 to 174) “The EEG data were originally sampled at 1000Hz and online-filtered between 0.1 Hz and 100 Hz during acquisition, with recordings referenced to the Fz electrode. For preprocessing, no additional filtering was applied. Baseline correction was performed by subtracting the mean signal during the 100 ms pre-stimulus interval from each trial and channel separately. We used already preprocessed data from 17 channels with labels beginning with “O” or “P” (O1, Oz, O2, PO7, PO3, POz, PO4, PO8, P7, P5, P3, P1, Pz, P2) ensuring full coverage of posterior regions typically involved in visual object processing. The epoched data were then down-sampled to 100 Hz.”

(b) Why was decoding accuracy used as the human RDM method rather than the EEG data themselves?

Thanks for your question! We would like to address why we used decoding accuracy for EEG RDMs rather than correlation. While fMRI RDMs are typically calculated using 1 minus correlation coefficient, decoding accuracy is more commonly used for EEG RDMs (Grootswager et al., 2017; Xie et al., 2020). The primary reason is that EEG signals are more susceptible to noise than fMRI data. Correlation-based methods are particularly sensitive to noise and may not reliably capture the functional differences between EEG patterns for different conditions. Decoding accuracy, by training classifiers to focus on task-relevant features, can effectively mitigate the impact of noisy signals and capture the representational difference between two conditions.

Grootswagers, T., Wardle, S. G., & Carlson, T. A. (2017). Decoding dynamic brain patterns from evoked responses: A tutorial on multivariate pattern analysis applied to time series neuroimaging data. Journal of Cognitive Neuroscience, 29(4), 677-697.

Xie, S., Kaiser, D., & Cichy, R. M. (2020). Visual imagery and perception share neural representations in the alpha frequency band. Current Biology, 30(13), 2621-2627.

We added this explanation to the manuscript:

(line 204 to 209) “Since EEG has a low SNR and includes rapid transient artifacts, Pearson correlations computed over very short time windows yield unstable dissimilarity estimates (Kappenman & Luck, 2010; Luck, 2014) and may thus fail to reliably detect differences between images. In contrast, decoding accuracy - by training classifiers to focus on task-relevant features - better mitigates noise and highlights representational differences.”

(c) How were the specific posterior electrodes selected?

The 17 posterior electrodes used in our analyses were pre-selected and provided in the THINGS EEG2 dataset, and corresponding to standard occipital and parietal sites based on the 10-10 EEG system. Specifically, we included all 17 electrodes with labels beginning with “O” or “P”, ensuring full coverage of posterior regions typically involved in visual object processing (Page 7).

(d) The specific layers should be named rather than the vague ("last visual")

Apologize for the confusing! We added more details about the layer information:

(line 255 to 257) “The early layer in ResNet refers to ResNet.maxpool layer, and the late layer in ResNet refers to ResNet.avgpool layer. The early layer in CLIP refers to CLIP.visual.avgpool layer, and the late layer in CLIP refers to CLIP.visual.attnpool layer.”

(line 420 to 434) “As shown in Figure 3F, the early layer representations of both ResNet and CLIP (ResNet.maxpool layer and CLIP.visual.avgpool) showed significant correlations with early EEG time windows (early layer of ResNet: 40-280ms, early layer of CLIP: 50-130ms and 160-260ms), while the late layers (ResNet.avgpool layer and CLIP.visual.attnpool layer) showed correlations extending into later time windows (late layer of ResNet: 80-300ms, late layer of CLIP: 70-300ms). Although there is substantial temporal overlap between early and late model layers, the overall pattern suggests a rough correspondence between model hierarchy and neural processing stages.

We further extended this analysis across intermediate layers of both ResNet and CLIP models (from early to late, ResNet: ResNet.maxpool, ResNet.layer1, ResNet.layer2, ResNet.layer3, ResNet.layer4, ResNet.avgpool; from early to late, CLIP: CLIP.visual.avgpool, CLIP.visual.layer1, CLIP.visual.layer2, CLIP.visual.layer3, CLIP.visual.layer4, CLIP.visual.attnpool).”

(e) p19: please change the reporting of t-statistics to standard APA format.

Thanks for the suggestion. We changed the reporting format accordingly:

(line 392 to 394) “The representation of real-word size had a significantly later peak latency than that of both retinal size, t(9)=4.30, p=.002, and real-world depth, t(9)=18.58, p<.001. And retinal size representation had a significantly later peak latency than real-world depth, t(9)=3.72, p=.005.”

(2) "early layer of CLIP: 50-130ms and 160-260ms, while the late layer representations of twoANNs were significantly correlated with later representations in the human brain (late layer of ResNet: 80-300ms, late layer of CLIP: 70-300ms)."This seems a little strong, given the large amount of overlap between these models.

We agree that our original wording may have overstated the distinction between early and late layers, given the substantial temporal overlap in their EEG correlations. We revised this sentence to soften the language to reflect the graded nature of the correspondence, and now describe the pattern as a general trend rather than a strict dissociation:

(line 420 to 427) “As shown in Figure 3F, the early layer representations of both ResNet and CLIP (ResNet.maxpool layer and CLIP.visual.avgpool) showed significant correlations with early EEG time windows (early layer of ResNet: 40-280ms, early layer of CLIP: 50-130ms and 160-260ms), while the late layers (ResNet.avgpool layer and CLIP.visual.attnpool layer) showed correlations extending into later time windows (late layer of ResNet: 80-300ms, late layer of CLIP: 70-300ms). Although there is substantial temporal overlap between early and late model layers, the overall pattern suggests a rough correspondence between model hierarchy and neural processing stages.”

(3) "Also, human brain representations showed a higher similarity to the early layer representation of the visual model (ResNet) than to the visual-semantic model (CLIP) at an early stage. "This has been previously reported by Greene & Hansen, 2020 J Neuro.

Thanks! We added this reference.

(4) "ANN (and Word2Vec) model RDMs"Why not just "model RDMs"? Might provide more clarity.

We chose to use the phrasing “ANN (and Word2Vec) model RDMs” to maintain clarity and avoid ambiguity. In the literature, the term “model RDMs” is sometimes used more broadly to include hypothesis-based feature spaces or conceptual models, and we wanted to clearly distinguish our use of RDMs derived from artificial neural networks and language models. Additionally, explicitly referring to ANN or Word2Vec RDMs improves clarity by specifying the model source of each RDM. We hope this clarification justifies our choice to retain the original phrasing for clarity.